# SEMI-SUPERVISED OFFLINE REINFORCEMENT LEARN-ING WITH ACTION-FREE TRAJECTORIES

## ABSTRACT

Natural agents can effectively learn from multiple data sources that differ in size, quality, and types of measurements. We study this heterogeneity in the context of offline reinforcement learning (RL) by introducing a new, practically motivated semi-supervised setting. Here, an agent has access to two sets of trajectories: labelled trajectories containing state, action, reward triplets at every timestep, along with unlabelled trajectories that contain only state and reward information. For this setting, we develop a simple meta-algorithmic pipeline that learns an inverse-dynamics model on the labelled data to obtain proxy-labels for the unlabelled data, followed by the use of any offline RL algorithm on the true and proxy-labelled trajectories. Empirically, we find this simple pipeline to be highly successful — on several D4RL benchmarks (Fu et al., 2020), certain offline RL algorithms can match the performance of variants trained on a fully labelled dataset even when we label only 10% trajectories from the low return regime. Finally, we perform a large-scale controlled empirical study investigating the interplay of data-centric properties of the labelled and unlabelled datasets, with algorithmic design choices (e.g., inverse dynamics, offline RL algorithm) to identify general trends and best practices for training RL agents on semi-supervised offline datasets.

## 1 INTRODUCTION

One of the key challenges with deploying reinforcement learning (RL) agents is its prohibitive sample complexity for real-world applications. Offline reinforcement learning (RL) can significantly reduce the sample complexity by exploiting logged demonstrations from auxiliary data sources (Levine et al., 2020). However, contrary to curated benchmarks in use today, the nature of offline demonstrations in the real world can be highly varied. For example, the demonstrations could be misaligned due to frequency mismatch (Burns et al., 2022), use of different sensors, actuators, or dynamics (Reed et al., 2022; Lee et al., 2022), or lacking partial state (Ghosh et al., 2022; Rafailov et al., 2021; Mazoure et al., 2021), or reward information (Yu et al., 2022). Successful offline RL in the real world requires embracing these heterogeneous aspects for maximal data efficiency, similar to learning in humans.

In this work, we propose a new semi-supervised setup for offline RL. Standard offline RL assumes trajectories to be sequences of observations, actions, and rewards. However, many data sources, such as videos or third-person demonstrations lack direct access to actions. Hence, we propose a semi-supervised setup, where an agent's offline dataset also consists of action-unlabelled trajectories in addition to the aforementioned (action-labelled) trajectories. Standard offline RL algorithms, such as Conservative Q Learning (CQL; Kumar et al. (2020)) or Decision Transformer (DT; Chen et al. (2021)), cannot directly operate on such unlabelled trajectories. At the same time, naively throwing out the unlabelled trajectories can be wasteful, especially when they have high returns. Our goal in this work is to enable compute and data efficient learning with additional action-unlabelled trajectory logs.

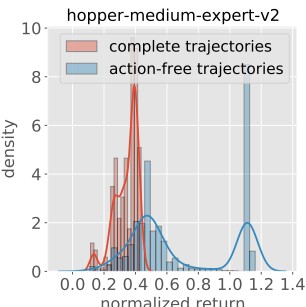

Figure 1.1: An example of the labelled and unlabelled data distributions.

Unlike traditional semi-supervised learning, our setup has a few key differences. First, we do not assume that the distribution of the labelled and unlabelled trajectories are necessarily identical. In realistic scenarios, we expect these to be different with unlabelled data

having higher returns than labelled data e.g., videos of a human professional are easier to obtain than installing actuators for continuous control tasks. We replicate such varied data quality setups in some of our experiments; Figure 1.1 shows an illustration of the difference in returns between the labelled and unlabelled dataset splits for the `hopper-medium-expert` D4RL dataset. Second, our end goal goes beyond labeling the actions in the unlabelled trajectories, but rather we intend to use the unlabelled data for learning a downstream policy that is better than the behavioral policies used for generating the offline datasets. Hence, there are two kinds of generalization challenges: generalizing from the labelled to the unlabelled data distribution and then going beyond the offline data distributions to get closer to the expert distribution. Regular offline RL is concerned only with the latter. Finally, we are mainly interested in the case where a significant majority of the trajectories in the offline dataset are unlabelled. One motivating example for this setup is learning from videos or third-person demos. There are tremendous amounts of internet videos that can be potentially used to train RL agents, yet they are without action labels and are of varying quality.

Our paper seeks to answer the following questions:

1. How can we utilize the unlabelled data for improving the performance of offline RL algorithms?
2. How does our performance vary as a function of data-centric properties, such as the size and return distributions of labelled and unlabelled datasets?
3. How do offline RL algorithms compare in this setup?

To answer these questions, we propose a meta-algorithmic pipeline to train policies in the semi-supervised setup described above. We call our pipeline **S**emi-**S**upervised **O**ffline **R**einforcement **L**earning (`SS-ORL`). `SS-ORL` contains three simple and scalable steps: (1) train a multi-transition inverse dynamics model on labelled data, which predicts actions based on transition sequences, (2) fill in proxy-actions for unlabelled data, and finally (3) train an offline RL agent on the combined dataset. Empirically, we instantiate `SS-ORL` with CQL (Kumar et al., 2020), DT (Chen et al., 2021), and TD3BC (Fujimoto & Gu, 2021) as the underlying offline RL algorithms respectively, and conduct experiments on the D4RL datasets (Fu et al., 2020). We highlight a few predominant trends from our experimental findings below:

1. Given low-quality labelled data, `SS-ORL` agents can exploit unlabelled data that contains high-quality trajectories and thus improve performance. The absolute performance of `SS-ORL` is close to or even matches that of the oracle agents, which have access to complete action information.
2. When the labelled data quality is high, utilizing unlabelled data does not bring significant benefits.
3. The choice of value vs. behavior cloning based methods can significantly affect performance in the semi-supervised setup. In our experiments, CQL and TD3BC are less sensitive to the missing actions compared to DT. They enjoy better absolute performance when the labelled data is of low quality, and their performance gap relative to the oracle agent is also smaller. See Appendix H for more details.

## 2 RELATED WORK

**Offline RL** The goal of offline RL is to learn effective policies from fixed datasets which are generated by unknown behavior policies. There are two main categories of model-free offline RL methods: value-based methods and behavior cloning (BC) based methods.

Value-based methods attempt to learn the value functions based on temporal difference (TD) updates. There is a line of work that aims to port existing off-policy value-based online RL methods to the offline setting, with various types of additional regularization components that encourage the learned policy to stay close to the behavior policy. Several representative techniques include specifically tailored policy parameterizations (Fujimoto et al., 2019; Ghasemipour et al., 2021), divergence-based regularization on the learned policy (Wu et al., 2019; Jaques et al., 2019; Kumar et al., 2019), and regularized value function estimation (Nachum et al., 2019; Kumar et al., 2020; Kostrikov et al., 2021a; Fujimoto & Gu, 2021; Kostrikov et al., 2021b).

Recently, a growing body of work has tried to formulate offline RL as a supervised learning problem (Chen et al., 2021; Janner et al., 2021; Emmons et al., 2021). Compared with the value-based methods, these methods enjoy several appealing properties including algorithmic simplicity and training stability. Generally speaking, these approaches can be viewed as conditional behavior cloning methods (Bain & Sammut, 1995), where the conditioning parameters are related information such

as goals or rewards. Similar to value-based methods, these can be extended to the online setup as well (Zheng et al., 2022) and demonstrate excellent performance in hybrid setups involving both offline data and online interactions.

**Semi-supervised Learning** Semi-supervised learning (SSL) is a sub-area of machine learning that studies approaches to train predictors from a small amount of labelled data combined with a large amount of unlabelled data. In supervised learning, predictors only learn from labelled data. However, labelled training examples often require human annotation efforts and are thus hard to obtain, whereas unlabelled data can be comparatively easy to collect. The research on semi-supervised learning spans several decades. One of the oldest SSL techniques, *self-training*, was originally proposed in the 1960s (Fralick, 1967). There, a predictor is first trained on the labelled data. Then, at each training round, according to certain selection criteria such as model uncertainty, a portion of the unlabelled data is annotated by the predictor and added into the training set for the next round. We refer the readers to Zhu (2005); Chapelle et al. (2006); Ouali et al. (2020); Van Engelen & Hoos (2020) for comprehensive literature surveys.

**Imitation Learning from Observations** There have been several works in imitation learning (IL) which do not assume access to the full set of actions, such as BCO (Torabi et al., 2018a), MoBILE (Kidambi et al., 2021), GAIfO (Torabi et al., 2018b) or third-person IL approaches (Stadie et al., 2017; Sharma et al., 2019). The recent work of Baker et al. (2022) also considered a setup where a small number of labelled actions are available in addition to a large unlabelled dataset. A key difference between our work and these is that the IL setup typically assumes that all trajectories are generated by an expert, unlike our offline setup. Further, some of these methods even permit reward-free interactions with the environment which is not possible in the offline setup.

**Learning from Videos** Closely related to IL from observations, several works (Schmeckpeper et al., 2020b;a) consider training agents with human video demonstrations, which are without action annotations. Distinct from out setup, in those works, the offline observational data (videos) are from a different embodiment. Moreover, the agents can interact with the environment, and can even collect reward information sometimes.

## 3   SEMI-SUPERVISED OFFLINE REINFORCEMENT LEARNING

**Preliminaries** We model our environment as a Markov decision process (MDP) (Bellman, 1957) denoted by $\langle \mathcal{S}, \mathcal{A}, p, P, R, \gamma \rangle$, where $\mathcal{S}$ is the state space, $\mathcal{A}$ is the action space, $p(s_1)$ is the distribution of the initial state, $P(s_{t+1}|s_t, a_t)$ is the transition probability distribution, $R(s_t, a_t)$ is the deterministic reward function, and $\gamma$ is the discount factor. At each timestep $t$, the agent observes a state $s_t \in \mathcal{S}$ and executes an action $a_t \in \mathcal{A}$. As a response, the environment moves the agent to the next state $s_{t+1} \sim P(\cdot|s_t, a_t)$, and also returns the agent a reward $r_t = R(s_t, a_t)$.

### 3.1   PROPOSED SETUP

We assume the agent has access to a static offline dataset $\mathcal{T}_{\text{offline}}$. The dataset consists of trajectories collected by certain unknown policies, which are not necessarily optimal. Let $\tau$ denote a trajectory and $|\tau|$ denote its length. We assume that all the trajectories in $\mathcal{T}_{\text{offline}}$ contain complete rewards and states. However, only a small subset of them contain action labels, while most of the trajectories are missing actions.

We are interested in learning a policy by leveraging the offline dataset without interacting with the environment. This setup is analogous to semi-supervised learning, where actions serve the role of *labels*. Hence, we also refer to the complete trajectories as *labelled* data (denoted by $\mathcal{T}_{\text{labelled}}$) and the action-free trajectories as *unlabelled* data (denoted by $\mathcal{T}_{\text{unlabelled}}$). Further, we assume the labelled data are sampled from a distribution $\mathbf{P}_{\text{labelled}}$ and the unlabelled data are sampled from $\mathbf{P}_{\text{unlabelled}}$. In general, the two distributions can be different. Practically, one case we are particularly interested in is when $\mathbf{P}_{\text{labelled}}$ generates low-to-moderate quality trajectories, whereas $\mathbf{P}_{\text{unlabelled}}$ generates trajectories of diverse qualities including ones with high returns.

Our setup shares some similarities with state-only imitation learning (Ijspeert et al., 2002; Bentivegna et al., 2002; Torabi et al., 2019) in the use of action-unlabelled trajectories. However, there are also some key differences. In state-only IL, the unlabelled demonstrations are from the same distribution as the labelled demonstrations and correspond to a near-optimal expert policy. In our setting, both

---

**Algorithm 1:** Semi-supervised offline RL (`SS-ORL`)

---

1 **Input:** trajectories $\mathcal{T}_{\text{labelled}}$ and $\mathcal{T}_{\text{unlabelled}}$, IDM transition size $k$, offline RL method `ORL`
   ```
   // train a stochastic multi-transition IDM using the labelled data
   ```
2 $\widehat{\theta} \leftarrow \operatorname{argmin}_\theta \mathbb{E}_{a_t, \mathbf{s}_{t-k:t+k+1} \sim \mathcal{T}_{\text{labelled}}} \left[ -\log \phi_\theta(a_t | \mathbf{s}_{t-k:t+k+1}) \right]$

   ```
   // fill in the proxy actions for the unlabelled data
   ```
3 $\mathcal{T}_{\text{proxy}} \leftarrow \varnothing$
4 **for** *each trajectory $\tau \in \mathcal{T}_{\text{unlabelled}}$* **do**
5   $\quad \widehat{a}_t \leftarrow$ mean of $\mathcal{N}\left( \mu_{\widehat{\theta}}(\mathbf{s}_{t-k:t+k+1}), \Sigma_{\widehat{\theta}}(\mathbf{s}_{t-k:t+k+1}) \right), t = 1, \ldots, |\tau|$
6   $\quad \tau_{\text{proxy}} \leftarrow \tau$ with proxy actions $\{\widehat{a}_t\}_{t=1}^{|\tau|}$ filled in
7   $\quad \mathcal{T}_{\text{proxy}} \leftarrow \mathcal{T}_{\text{proxy}} \bigcup \{\tau_{\text{proxy}}\}$

   ```
   // train an offline RL agent using the combined data
   ```
8 $\pi \leftarrow$ policy obtained by training `ORL` using dataset $\mathcal{T}_{\text{labelled}} \bigcup \mathcal{T}_{\text{proxy}}$
9 **Output:** $\pi$

---

$\mathbf{P}_{\text{labelled}}$ and $\mathbf{P}_{\text{unlabelled}}$ can be different from each other and also from the expert policy. Further, many state-only imitation learning algorithms (e.g., Gupta et al. (2017); Torabi et al. (2018a;b); Liu et al. (2018); Sermanet et al. (2018)), similar to their original counterparts (e.g., Ho & Ermon (2016); Kim et al. (2020)), permit (reward-free) interactions with the environments. This is not possible in our proposed offline semi-supervised setup where the agents are only provided with $\mathcal{T}_{\text{labelled}}$ and $\mathcal{T}_{\text{unlabelled}}$.

### 3.2 TRAINING PIPELINE

RL policies trained on low to moderate quality offline trajectories are often sub-optimal, as many of the trajectories might not have high return and only cover a limited part of the state space. Our goal is to find a way to combine the action labelled trajectories and the unlabelled action-free trajectories, so that the offline agent can exploit structures in the unlabelled data to improve performance.

One natural strategy is to fill in *proxy actions* for those unlabelled trajectories, and use the annotated data together with the labelled data as a whole to train an offline RL agent. Since we assume both the labelled and unlabelled trajectories contain the states, we can train an inverse dynamics model (IDM) $\phi$ that predicts actions using the states. Once we obtain the IDM, we use it to generate the proxy actions for the unlabelled trajectories. Finally, we combine those proxy-labelled trajectories with the labelled trajectories, and train an agent using the offline RL algorithm of choice. In particular, we propose a stochastic multi-transition IDM (see Section 3.3), which is favored by our experiments. Our meta-algorithmic pipeline is summarized in Algorithm 1.

**Remarks.** The annotation process, which involves training an IDM on the labelled data and generating proxy actions for the unlabelled trajectories, is similar to one step of *self-training* (Fralick, 1967). A key difference is that in self-training, the predictor is trained in multiple rounds. Once an initial predictor is trained, it is used for obtaining annotations on the unlabelled dataset. Then, a subset of annotated data is selected according to certain criteria, and added into the training set for the next round. As opposed to self-training, we do not retrain the IDM but directly move to the next stage, where we train the agent using the combined data.

There are a few reasons that we do not employ self-training for IDM. First, it is computationally expensive to execute multiple rounds of training. More importantly, our end goal is to obtain a downstream policy with improved performance via utilizing the proxy-labelled data. One commonly used data selection criterion for self-training is based on the model uncertainty. There, one adds the proxy-labelled data with sufficiently low predictive uncertainty into the training set for the next round. However, we empirically found that such an uncertainty based augmentation strategy did not improve the performance of `SS-ORL` agents. See Section 4.3 and Appendix F for the experiment details.

### 3.3 STOCHASTIC MULTI-TRANSITION INVERSE DYNAMIC MODEL

In past work (Pathak et al., 2017), the IDM typically maps two subsequent states $(s_t, s_{t+1})$ to $a_t$. We introduce a multi-transition IDM that predicts $a_t$ using both transitions before and after timestep $t$, which we found works better empirically. More precisely, our inverse dynamic model which predicts $a_t$ using $2k + 1$ transitions, including the current transition $(s_t, s_{t+1})$, the previous $k$ transitions that leads to $s_t$, and the next $k$ transitions starting from $s_{t+1}$. We call $k$ the transition size parameter.

Let $\mathbf{s}_{t-k:t+k+1}$ denote the sequence $s_{\min(0,t-k)}, \ldots, s_t, s_{t+1}, \ldots, s_{\max(|\tau|,t+k+1)})$. Specifically, we model the distribution of $a_t$ as a multivariate Gaussian distribution with a diagonal covariance matrix:

$$a_t \sim \mathcal{N}\big(\mu_\theta(\mathbf{s}_{t-k:t+k+1}), \Sigma_\theta(\mathbf{s}_{t-k:t+k+1})\big). \tag{1}$$

Let $\phi_\theta(a_t|\mathbf{s}_{t-k:t+k+1})$ be the probability density function of $\mathcal{N}\big(\mu_\theta(\mathbf{s}_{t-k:t+k+1}), \Sigma_\theta(\mathbf{s}_{t-k:t+k+1})\big)$. Given the labelled trajectories $\mathcal{T}_{\text{labelled}}$, we minimize the negative log-likelihood loss $\mathbb{E}_{a_t,\mathbf{s}_{t-k:t+k+1}\sim\mathcal{T}_{\text{labelled}}}\big[-\log\phi_\theta(a_t|\mathbf{s}_{t-k:t+k+1})\big]$. Note that the standard IDM which predicts $a_t$ from $(s_t, s_{t+1})$ under the $\ell_2$ loss, is a special case subsumed by our model: it is equivalent to the case $k = 0$ and the diagonal entries of $\Sigma_\theta$ (i.e., the variances of each action dimension) are all the same. Choosing $k > 0$ allows us to account for non-Markovian behaviour policies and partially observable MDP (POMDP), see Appendix E.1. For all the experiments in this paper, we use $k = 1$. We ablate this design choice in Section 4.3.

## 4 EXPERIMENTS

Our experiments aim to answer three primary questions:

1. Can SS-ORL closely track or even match the performance for fully supervised offline reinforcement learning, when only a small subset of trajectories are labelled?
2. How does the performance of SS-ORL vary as a function of the size and qualities of the labelled and unlabelled datasets?
3. Do different offline RL methods respond differently under varying setups of data size and qualities?

To answer these questions, we focus on three Gym locomotion tasks hopper, walker, and halfcheetah, and we use the v2 medium-expert, medium and medium-replay datasets[1] from the D4RL benchmark (Fu et al., 2020). We address the first question in Section 4.1 and the other two in Section 4.2, respectively. Finally, we discuss the design choices for SS-ORL in Section 4.3.

### 4.1 BENCHMARKING

**Data Setup** For a given offline dataset, we subsample $10\%$ of the total trajectories from the dataset, whose returns are from the bottom $q\%$, $10 \leqslant q \leqslant 100$. We keep the actions for those trajectories, and discard the actions for the rest. We call this setup the *coupled* setup, since $\mathbf{P}_{\text{labelled}}$ and $\mathbf{P}_{\text{unlabelled}}$ will change simultaneously when we vary the value of $q$. When $q = 100$, we are uniformly sampling the trajectories and we have $\mathbf{P}_{\text{labelled}} = \mathbf{P}_{\text{unlabelled}}$. Under this setup, we always have $10\%$ trajectories labelled and $90\%$ unlabelled, and the total amount of data used later for training the offline RL agent is the original offline dataset size. This allows us to easily compare our results with results under the standard, fully labelled setup. In Section 4.2, we shall decouple the distributions $\mathbf{P}_{\text{labelled}}$ and $\mathbf{P}_{\text{unlabelled}}$ for a thorough understanding of their individual influences.

**Inverse Dynamic Model** We train an IDM as described in Section 3 with parameter $k = 1$. In other words, the IDM predicts $a_t$ using 3 consecutive transitions: $(s_{t-1}, s_t, s_{t+1}, s_{t+2})$. The mean and the covariance matrix are predicted by two independent multilayer perceptrons (MLPs), each contains two hidden layers and 1024 hidden units per layer. To prevent overfitting, we randomly sample $10\%$ of the labelled trajectories as the validation set, and use the IDM that yields the best validation error within 100k training iterations.

**Offline RL Methods** We instantiate Algorithm 1 with DT, CQL and TD3BC (Fujimoto & Gu, 2021) and test their performances. Among these methods, DT is a recently proposed conditional behavior cloning (BC) method that uses sequence modeling tools to model the trajectories; CQL is a representative value-based offline RL method; and TD3BC is a hybrid method which adds a BC term to regularize the Q-learning updates. We refer to those instantiations as SS-DT, SS-CQL and SS-TD3BC, respectively. We defer the implementation details to Appendix A.

**Results** We compare the performance of those SS-ORL agents with corresponding *baseline* and *oracle* agents. The baseline agents are trained on the labelled trajectories only, and the oracle agents are trained on the full offline dataset with action labels. Intuitively, the performances of the baseline and the oracle agents can be considered as the (estimated) lower and upper bounds for the performance of the SS-ORL agents. For each method, we train 5 instances under different seeds, and

---

[1]Due to the space limit, the results on medium and medium-replay datasets are deferred to Appendix C.

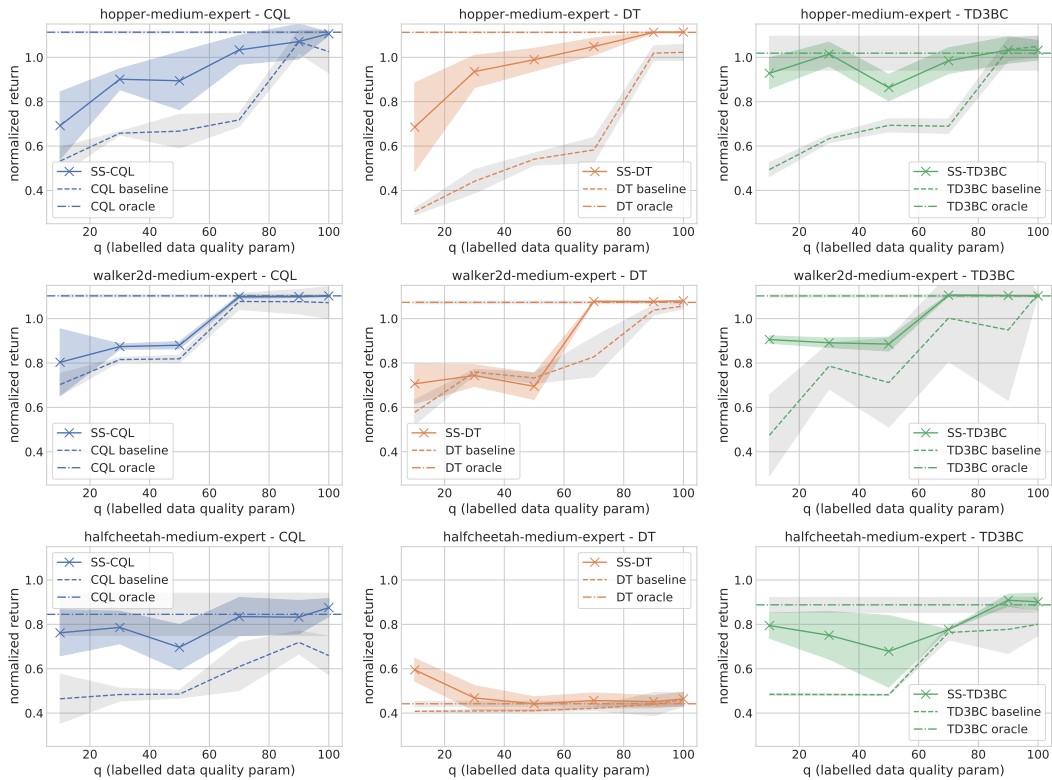

Figure 4.1: The return (average and standard deviation) of `SS-ORL` agents trained on the D4RL `medium-expert` datasets. The `SS-ORL` agents are able to make use of the unlabelled data to improve their performance, and can even match the performance of oracle agents trained on the fully labelled offline datasets.

for each instance we run 30 evaluation trajectories. We report the average return and the standard deviation after 200k iterations. Figure 4.1 plots the results on `medium-expert` datasets. For all the three environments and all the three offline RL methods, the `SS-ORL` agents improve upon the baselines. Remarkably, even when the labelled data quality is low, the `SS-ORL` agents are able to obtain decent returns. For example, when $q = 10$, i.e., the labelled trajectories are the bottom $10\%$ trajectories, the average return obtained by `SS-TD3BC` is $0.93$, $0.91$ and $0.79$ for `hopper`, `walker` and `halfcheetah`. On average, this is $87.4\%$ relative to the oracle performance ($1.02$, $1.1$ and $0.89$). As the value $q$ increases, the labelled data quality increases and the distributions $\mathbf{P}_{\text{labelled}}$ and $\mathbf{P}_{\text{unlabelled}}$ are getting closer. The performance of the `SS-ORL` agents also keeps increasing and finally matches the performance of the oracle agents. Similar observations can be found in the results of `medium` and `medium-replay` datasets, see Figure C.1 and C.2. We found relatively suboptimal results for DT on `halfcheetah` in all cases, consistent with prior results in Zheng et al. (2022).

## 4.2 ABLATION STUDY

We conduct experiments to understand the semi-supervised approach from the perspective of both datasets and learning algorithms. For a systematic study, we depart from the coupled setup in Section 4.1 and consider a decoupling of the labelled data distributions $\mathbf{P}_{\text{labelled}}$ and the unlabelled data distribution $\mathbf{P}_{\text{unlabelled}}$. We first vary the quality of the labelled and unlabelled trajectories, and examine how the final performance of those `SS-ORL` agents changes. Next, we vary the size of the labelled and unlabelled trajectories and investigate their influences. To understand how the value-based methods and the BC methods will potentially react differently under these data setups. we report the results of `SS-CQL` and `SS-DT` for the aforementioned setups. Last, we ablate the design choice of the transition size $k$ for the proposed IDM. For all the experiments we present, the results are aggregated over 5 instances with different seeds.

**Quality of Unlabelled Data**  We divide the trajectories from the `hopper-medium-expert` dataset into 3 groups, which consist of trajectories whose returns are the bottom 0% to 33%, 33% to 67%, and

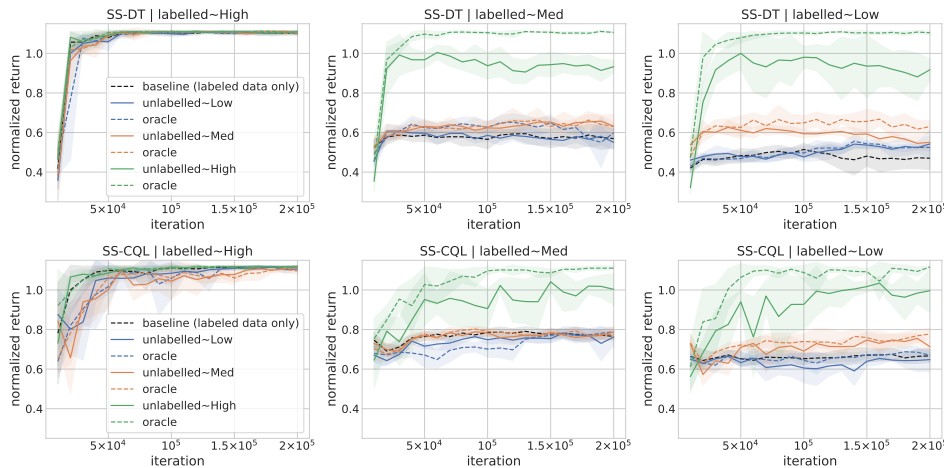

(a) We fix the labelled data quality and vary the unlabelled data quality. When the labelled data quality is low or moderate, `SS-ORL` can significantly improve the performance upon the baselines by utilizing high quality unlabelled data.

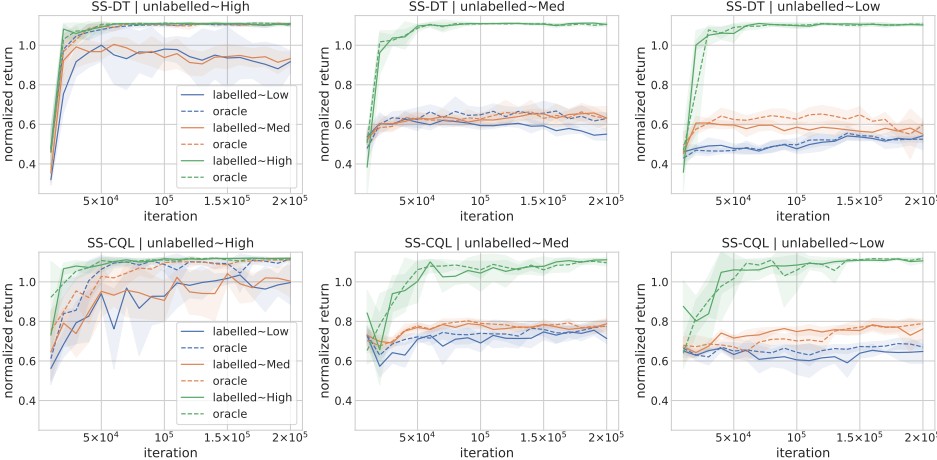

(b) We fix the unlabelled data quality and vary the labelled data quality. The performance of `SS-ORL` improves as the labelled data quality increases.

Figure 4.2: The return (average and standard deviation) of `SS-DT` and `SS-CQL` agents trained on the `hopper-medium-expert` dataset, when the qualities of the labelled and unlabelled data vary. Both the sizes of the labelled and unlabelled data are $10\%$ of the offline dataset size.

67% to 100%, respectively. We refer to them `Low`, `Medium`, and `High` quality groups. In particular, the `High` group contains trajectories generated by the expert agents (Fu et al., 2020). As before, we report the performance of DT and CQL agents trained on the labelled data only as the baselines. We also report the results under the *oracle* mode, where we fill in the unlabelled trajectories with the true actions, and combine them with the labelled trajectories to train offline RL agents.

We first report the performance of `SS-DT` when the labelled data is sampled from the `High` group, and the unlabelled data are sampled from `Low`, `Med`, and `High` groups, respectively. Both the size of the labelled and unlabelled trajectories are $10\%$ of the total offline dataset size. The top left panel of Figure 4.2a plots the results. Clearly, when the labelled data quality is high, training on the labelled data only is sufficient to achieve the expert performance, and adding unlabelled data does not bring extra benefits. We repeat the same experiment when the labelled data is sampled from `Medium` and `Low`, see the top middle and top right panels of Figure 4.2a. For those cases, adding unlabelled data with the higher or the same quality[2] improves the performance, whereas the lower quality unlabelled

---

[2]When the labelled and unlabelled data are sampled from the same quality group, we are simply adding more data from the same distribution.

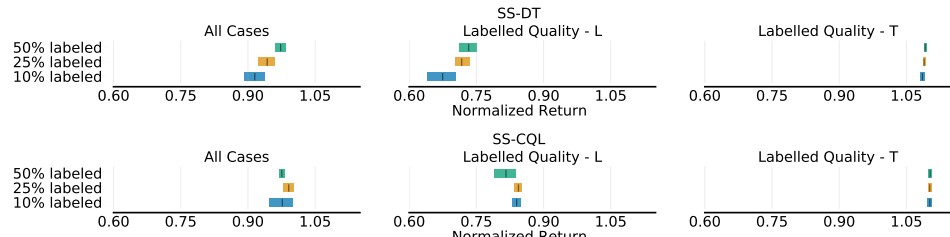

(a) We fix the unlabelled data size to be $10\%$ of the offline dataset size, and vary the labelled data size.

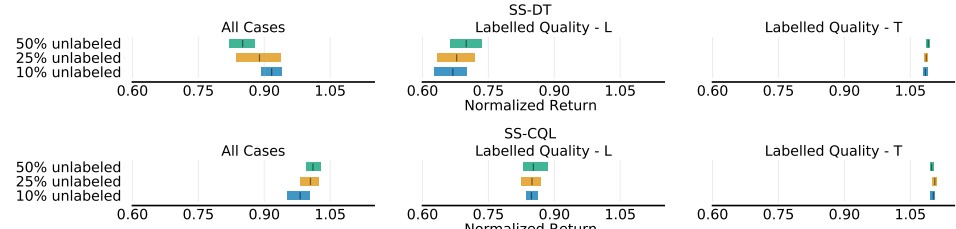

(b) We fix the labelled data size to be $10\%$ of the offline dataset size, and vary the unlabelled data size.

Figure 4.3: The $95\%$ stratified bootstrap CIs of the interquartile mean returns of `SS-DT` and `SS-CQL` when the sizes of the labelled and unlabelled data change. `SS-CQL` is almost insensitive to the changes, whereas `SS-DT` is more sensitive when the labelled data quality is low.

data is not significantly helpful. The performance of `SS-CQL` follows the same trends, see the bottom panels of Figure 4.2a.

To summarize, the experiments provide strong evidence that when the labelled data is of low or moderate quality, `SS-ORL` is capable to exploit the high quality unlabelled data and remarkably boosts the performance compared with the baselines. The resulting performance is close to that of the oracle agent, and is often optimal (at least 1) or near-optimal (close to 1).

**Quality of Labelled Data**  Similarly, we fixed the unlabelled data quality and vary the quality of the labelled data. Figure 4.2b shows the results. For both `SS-DT` and `SS-CQL`, increasing the labelled data quality raises the performance for all the cases.

**Size of Labelled Data**  We train `SS-ORL` agents where we fix the number of unlabelled trajectories to be $10\%$ of the total number of offline trajectories, and vary the number of labelled trajectories as $10\%$, $25\%$, and $50\%$ of the total size. Similar to the above experiments, we consider four data quality setups, where the labelled and unlabelled trajectories are sampled from the bottom half (denoted by `L`) and top half (denoted by `T`) trajectories, respectively. We consider both `hopper-medium-expert` and `walker-medium-expert` datasets. To take account of different environments and data setups, we report the $95\%$ stratified bootstrap confidence intervals (CIs) of the interquartile mean[3] of the return for all these cases and training instances (Agarwal et al., 2021). We use 50000 bootstrap replications to generate the CIs. Compared with some other statistics like the mean or the median, the IQM is robust to outliers and also a good representative of the overall performance. The stratified bootstrapping is a handy tool to obtain CIs with descent coverage rate, even if one only have a small number of training instances per setup. We refer the readers to Agarwal et al. (2021) for the complete introduction.

Figure 4.3a plots the confidence intervals when we consider all four quality setups, or when the labelled data quality is low or high, respectively. We found that `SS-DT` and `SS-CQL` respond slightly differently. Overall, `SS-CQL` is almost immune to changes in the size of the labelled data, as is `SS-DT` when the labelled data quality is high. However, `SS-DT`'s performance moderately increases as the labelled size grows when the labelled data quality is low. More detailed results, including the plots of the evaluation curves, CIs of the mean and the median, can be found in Appendix D.

**Size of Unlabelled Data**  As before, we vary the number of labelled data size with the unlabelled data size fixed, and report the $95\%$ stratified bootstrap CIs in Figure 4.3b. Similarly, `SS-CQL` is almost insensitive whereas `SS-DT` is sensitive when the labelled quality is low.

---

[3]The interquartile mean of a list of sorted numbers is the mean of the middle $50\%$ numbers.

**Value-based vs. Conditional BC** As discussed above, `SS-CQL` is insensitive to the data size changes, whereas `SS-DT` is more responsive when the labelled data quality is low. Regarding the data quality, we are mostly interested in the scenarios where the labelled data quality is low or moderate, see the red and blue curves of Figure 4.2b. In that regime, if the unlabelled data quality is high (the left column), the distribution shift from the labelled data to the unlabelled data is challenging to handle, and the proxy-actions predicted by the IDM will be less accurate. There, the absolute performance of `SS-CQL` is slightly better than `SS-DT`, with smaller performance gaps compared to the oracle agents. If the unlabelled data quality is moderate or low (the middle and right columns), `SS-CQL` clearly outperforms `SS-DT`. Both observations suggest that `SS-CQL` is less sensitive to the action quality.

### 4.3 DESIGN CHOICES

**Transition Size** $k$ **of the IDM** We train `SS-TD3BC`, `SS-CQL` and `SS-DT` agents on the `hopper-medium-expert` dataset, under the coupled setup as in Section 4.1. We consider 6 different values of $q$: 10, 30, 50, 70, 90 and 100. For all the 18 setups (3 `SS-ORL` agents and 6 different $q$ values), we train the agents using the multi-transition IDM with $k = 0, 1, 2$ respectively. As in the previous section, Figure E.1 plots the 95% stratified bootstrapped CIs for the IQM return across all the setups and training instances, which are generated by 50000 bootstrap replications. The results favors the choice $k = 1$. See Appendix E for more experiment details, such as the average return for each setup and the CIs for the return mean.

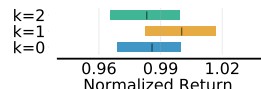

Figure 4.4: The 95% stratified bootstrap CIs of the IQM returns obtained by `SS-ORL` agents using different values of $k$.

**Data Augmentation Strategy** As discussed in Section 3.2, we consider variants of `SS-TD3BC` and `SS-DT` using uncertainty based data augmentation. Following Lakshminarayanan et al. (2017), we train an ensemble of 3 independent IDMs on $\mathcal{T}_{\text{labelled}}$. We generate the proxy actions for the unlabelled trajectories using the combined model, and also estimate the predictive uncertainties. We then only add proxy-labelled data whose uncertainties are below $p\%$ to the final RL training dataset. Specifically, we test 4 values of $p$: 25, 50, 75 and 95. We compare the results with standard `SS-TD3BC` and `SS-DT` where all the proxy-labelled data are added into the final RL training dataset. Again, we consider both the `hopper-medium-expert` and `walker-medium-expert` datasets and use the coupled setup under with 4 different $q$ values: 10, 30, 68, 75 for `hopper-medium-expert` and 10, 30, 54, 60 for `walker-medium-expert`. Figure 4.5 plots the

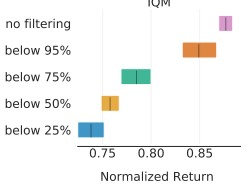

Figure 4.5: The 95% stratified bootstrap CIs of the IQM returns obtained by `SS-ORL` agents with different data augmentation strategies.

95% stratified bootstrap CIs of the IQM return across all the setups. Adding all the proxy-labelled data without filtering outperforms uncertainty based data augmentation; see Appendix F for more details. Intuitively, to make use of the unlabelled data, most of the SSL pipelines would assume $\mathbf{P}_{\text{labelled}}$ and $\mathbf{P}_{\text{unlabelled}}$ are similar or even the same (Chapelle et al., 2006). This is not the case in our setup, where $\mathbf{P}_{\text{labelled}}$ only generates low return trajectories, and all the high return ones come from $\mathbf{P}_{\text{unlabelled}}$. It remains an open question if self-training with the uncertainty based selection rule can help us generalize to high return trajectories.

## 5 DISCUSSION

We proposed a novel setup for offline RL where the trajectories do not have all of the action information, for which we have introduced a semi-supervised meta-algorithmic pipeline. Our experiments identified key properties that enable the agents to learn from unlabelled data and show that near-optimal learning can be done with only 10% of the actions labelled for low-to-moderate quality trajectories. It would be interesting to study other heterogeneous data setups for offline RL in the future, including reward-free or pure state-only settings.

This work is a step towards a broader goal of empowering robotic systems with the ability to extract meaningful knowledge from copious and ever-growing amounts of unlabelled demonstration data. Beyond simply not having the action labels, many trajectories may be from *different* robotic systems or tasks and therefore are not directly transferable to the system and task at hand. As we continue building robotic systems to leverage these forms of auxiliary knowledge, we expect that weakly-supervised learning paradigms such as the one explored in this work will be useful.

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

# A    EXPERIMENT DETAILS

In this section, we provide more details about our experiments. For all the offline RL methods we consider, we use our own implementations adopted from the following codebases:

DT [https://github.com/facebookresearch/online-dt](https://github.com/facebookresearch/online-dt)
TD3BC [https://github.com/sfujim/TD3_BC](https://github.com/sfujim/TD3_BC)
CQL [https://github.com/scottemmons/youngs-cql](https://github.com/scottemmons/youngs-cql)

We use the stochastic DT proposed by Zheng et al. (2022). For offline RL, its performance is similar to the deterministic DT (Chen et al., 2021). The policy parameter is optimized by the LAMB optimizer (You et al., 2019) with $\varepsilon = 10^{-8}$. The log-temperature parameter is optimized by the Adam optimzier (Kingma & Ba, 2014). The architecture and other hyperparameters are listed in Tabel A.1. For TD3BC, we optimize both the critic and actor parameters by the Adam optimizer. The complete hyperparameters are listed in Table A.2. For CQL, we also use the Adam optimizer to optimize the critic, actor and the log-temperature parameters. The architecture of critic and actor networks and the other hyperparameters are listed in Table A.3.

We use batch size 256 and context length 20 for DT, where each batch contains 5120 states. Correspondingly, we use batch size 5120 for CQL and TD3BC.

| Hyperparameter | Value |
|---|---|
| number of layers | 4 |
| number of attention heads | 4 |
| embedding dimension | 512 |
| context length | 20 |
| dropout | 0.1 |
| activation function | relu |
| batch size | 256 |
| learning rate for policy | 0.0001 |
| weight decay for policy | 0.001 |
| learning rate for log-temperature | 0.0001 |
| gradient norm clip | 0.25 |
| learning rate warmup | linear warmup for $10^4$ steps |
| target entropy | $-\dim(\mathcal{A})$ |
| evaluation return-to-go | 3600 Hopper |
|  | 5000 Walker |
|  | 6000 HalfCheetah |

Table A.1: The hyperparameters used for DT.

| Hyperparameter | Value |
|---|---|
| discount factor | 0.99 |
| target update rate | 0.005 |
| policy noise | 0.2 |
| policy noise clipping | $(-0.5, 0.5)$ |
| policy update frequency | 2 |
| critic learning rate | 0.0003 |
| critic hidden dim | 256 |
| critic hidden layers | 2 |
| actor learning rate | 0.0003 |
| actor hidden dim | 256 |
| actor hidden layers | 2 |
| activation function | ReLU |
| regularization parameter $\alpha$ | 2.5 |

Table A.2: The hyperparameters used for TD3BC.

| Hyperparameter | Value |
|---|---|
| discount factor | 0.99 |
| target update rate | 0.005 |
| critic learning rate | 0.0003 |
| critic hidden dim | 256 |
| critic hidden layers | 3 |
| actor learning rate | 0.0001 |
| actor hidden dim | 256 |
| actor hidden layers | 3 |
| log-temperature learning rate | 0.0003 |
| activation function | ReLU |
| number of sampled actions | 10 |
| target entropy | $-\dim(\mathcal{A})$ |
| minimum Q weight value | 5 |
| Lagrange | False |
| Importance Sampling | True |

Table A.3: The hyperparameters used for CQL.

## B THE RETURN DISTRIBUTIONS OF THE D4RL DATASETS

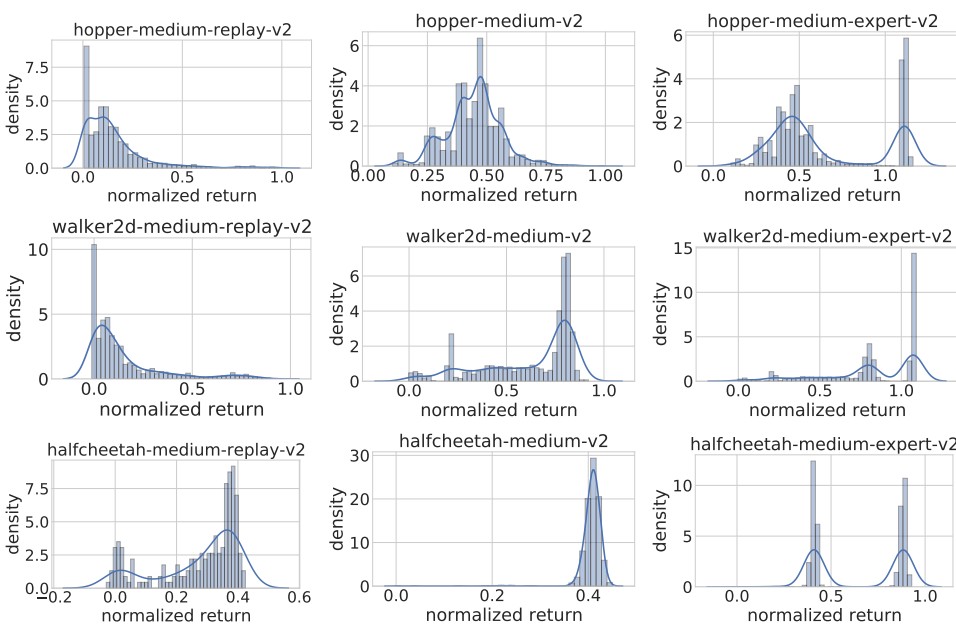

Figure B.1: The distributions of the normalized returns of the D4RL datasets.

## C ADDITIONAL EXPERIMENTS UNDER THE COUPLED SETUP

We conduct experiments on the `medium` and `medium-replay` datasets of D4RL benchmark, using the same setup as in Section 4.1. Figure C.1 and C.2 reports the results. The general trend is as the same as that in Figure 4.1. We note that the results on the `halfcheetah-medium` dataset are less informative than the others. This is because the data distributions of `halfcheetah-medium` is very concentrated, similar to a Gaussian distribution with small variance, see Figure B.1. In such a case, varying the value of $q$ does not drastically change the labelled data distribution.

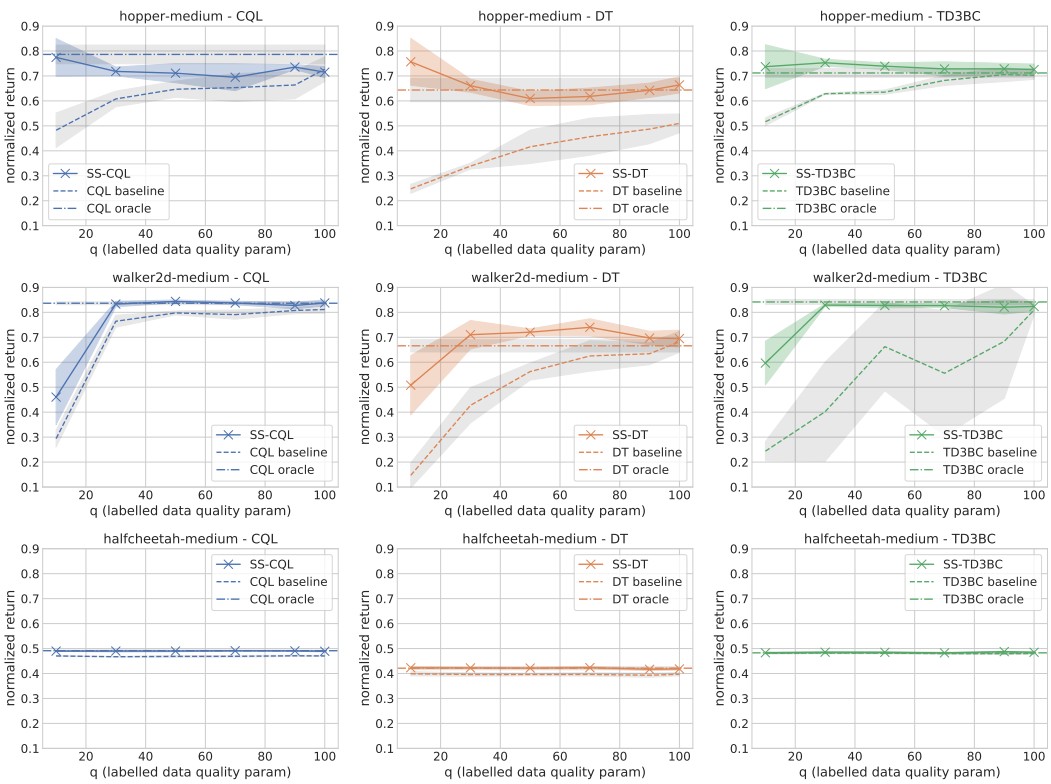

Figure C.1: The return (average and standard deviation) of `SS-ORL` agents trained on the D4RL `medium` datasets.

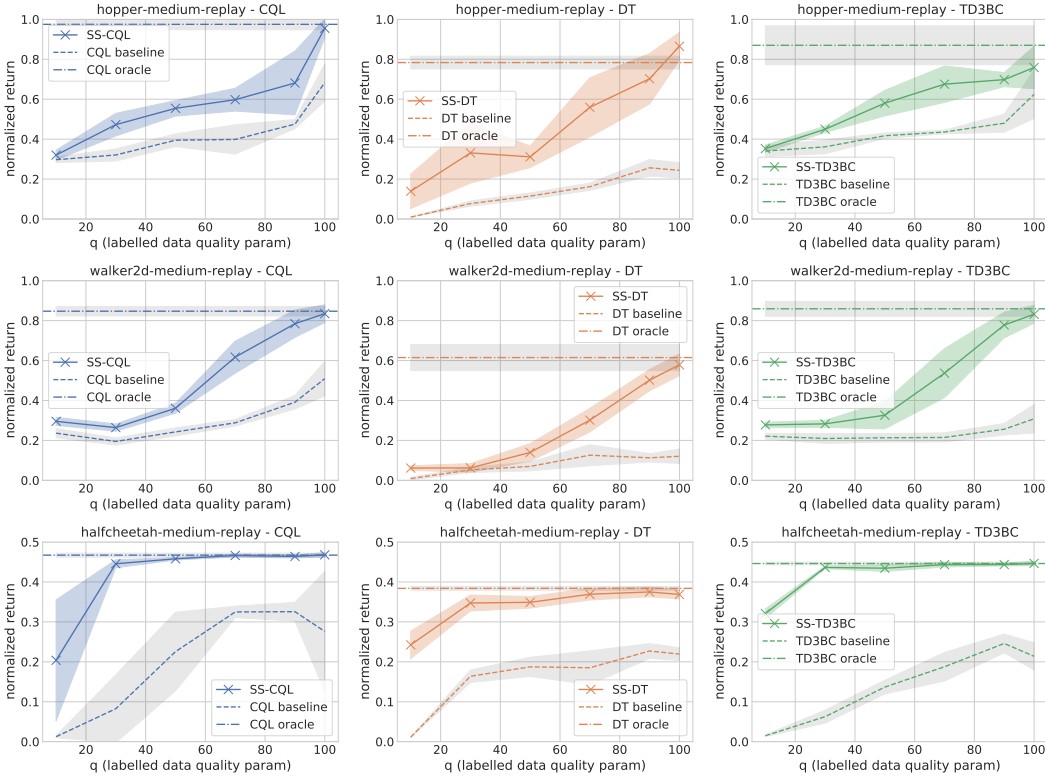

Figure C.2: The return (average and standard deviation) of `SS-ORL` agents on the D4RL `medium-replay` datasets.

| dataset | q=10 | q=30 | q=50 | q=70 | q=90 | q=100 |
|---|---|---|---|---|---|---|
| hopper-medium-replay | 0.007 | 0.022 | 0.05 | 0.074 | 0.109 | 0.149 |
| walker2d-medium-replay | -0.002 | 0.005 | 0.023 | 0.048 | 0.087 | 0.156 |
| halfcheetah-medium-replay | 0.001 | 0.092 | 0.179 | 0.202 | 0.269 | 0.275 |
| hopper-medium | 0.231 | 0.310 | 0.355 | 0.388 | 0.418 | 0.443 |
| walker2d-medium | 0.135 | 0.287 | 0.44 | 0.557 | 0.599 | 0.618 |
| halfcheetah-medium | 0.361 | 0.383 | 0.397 | 0.396 | 0.406 | 0.405 |
| hopper-medium-expert | 0.252 | 0.341 | 0.394 | 0.451 | 0.594 | 0.645 |
| walker2d-medium-expert | 0.201 | 0.469 | 0.605 | 0.732 | 0.791 | 0.827 |
| halfcheetah-medium-expert | 0.377 | 0.397 | 0.405 | 0.537 | 0.604 | 0.638 |

Table C.1: The average return of the labelled trajectories as $q$ increases in our experiments. Results aggreagted over 5 seeds.

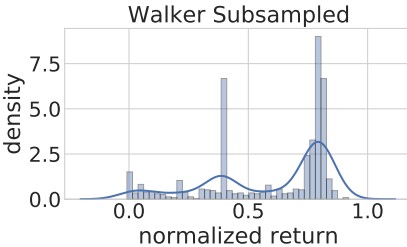

Figure C.3: The density of a randomly subsampled dataset of the `walker` environment.

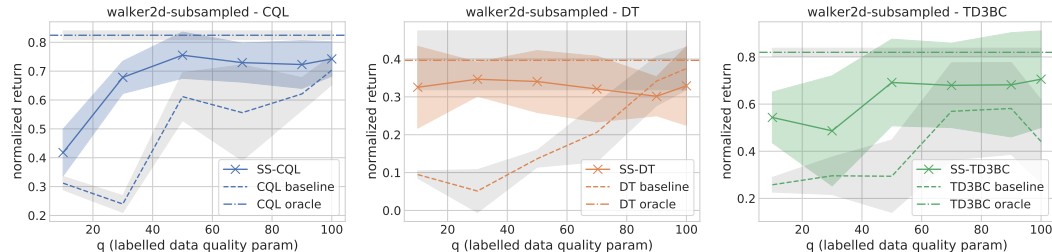

Figure C.4: The return (average and standard deviation) of `SS-ORL` agents on the subsampled dataset.

One may notice that for the `hopper-medium-replay` and `walker-medium-replay` datasets, `SS-ORL` does not catch up with the oracle as quickly as on the other datasets as $q$ increases. Our intuition is that the return distributions of these two datasets concentrate on extremely low values, as shown in Figure B.1. In our experiments, the labelled trajectories for those two datasets have average return small than 0.1 even when $q = 70$. In contrast, the return distributions of the other datasets concentrate on larger values. For the `halfcheetah-medium-replay` and all the `medium` and `medium-expert` datasets, increasing the value of $q$ will greatly change the returns of labelled trajectories, see Table C.1.

To demonstrate the performance of `SS-ORL` on dataset with a more wide return distribution, we consider a subsampled dataset for the `walker` environment generated as follows.

1. Combine the `walker-medium-replay` and `walker-medium` datasets.

2. Let $R_{\min}$ and $R_{\max}$ denote the minimum and maximum return in the dataset. We divide the trajectories into 40 bins, where the maximum returns within each bin are linear spaced between $R_{\min}$ and $R_{\max}$. Let $n_i$ be the number trajectories in bin $i$.

3. We randomly sample 1000 trajectories. To sample a trajectory, we first first sample a bin $i \in [1, \ldots, 40]$ with weights proportional to $1/n_i$, then sample a trajectory uniformly at random from the sampled bin.

Figure C.3 plots the return distribution of the subsampled dataset. It is wide and has 3 modes. We run the same experiments as before on this subsampled dataset, and Figure C.4 plots the results. We can see that `SS-ORL` methods can catch up with the oracle agents even when $q$ is small.

# D  INFLUENCES OF THE LABELLED AND UNLABELLED DATA SIZE

Figure D.1 plots the average return of `SS-DT` and `SS-CQL` when we fix the number of unlabelled trajectories and vary the number of labelled trajectories. We found that there is a bad seed for `SS-CQL` when both labelled and unlabelled trajectories are sampled from `L` and the size of labelled data is 10%, so that the result there the bottom right panel) exhibits large variance. Correspondingly, Figure D.2 plots the 95% stratified bootstrap CIs for the median, mean, interquartile mean, and the optimality gap of the return of `SS-DT` and `SS-CQL`.

Similarly, Figure D.3 and D.4 plots the results when we vary the number of unlabelled trajectories, while the number of unlabelled ones is fixed.

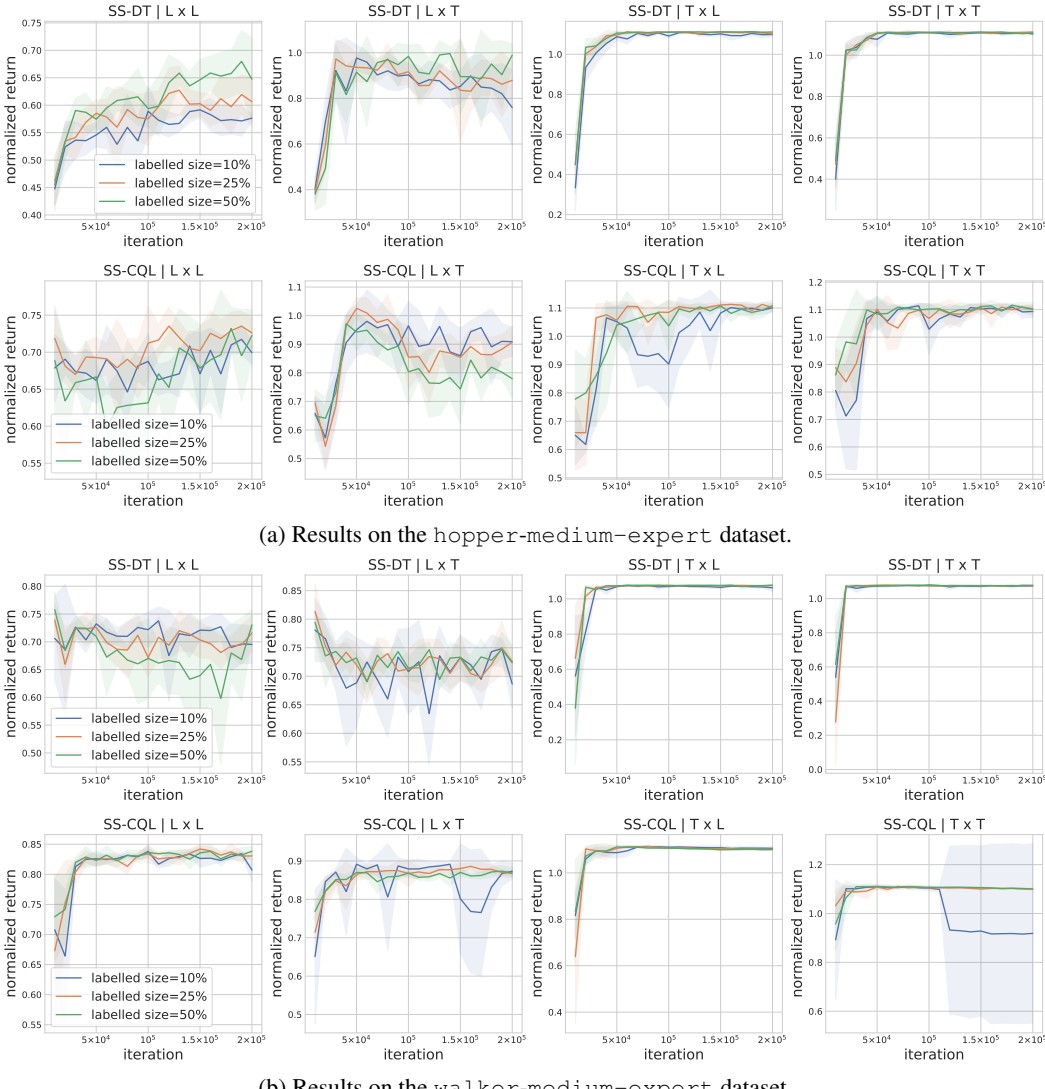

(a) Results on the `hopper-medium-expert` dataset.

(b) Results on the `walker-medium-expert` dataset.

Figure D.1: The return (average and standard deviation) of `SS-DT` and `SS-CQL` agents trained on the `medium-expert` datasets with different sizes of labelled data. The unlabelled data size is fixed to be 10% of the offline dataset size. Results aggregated over 5 instances with different seeds.

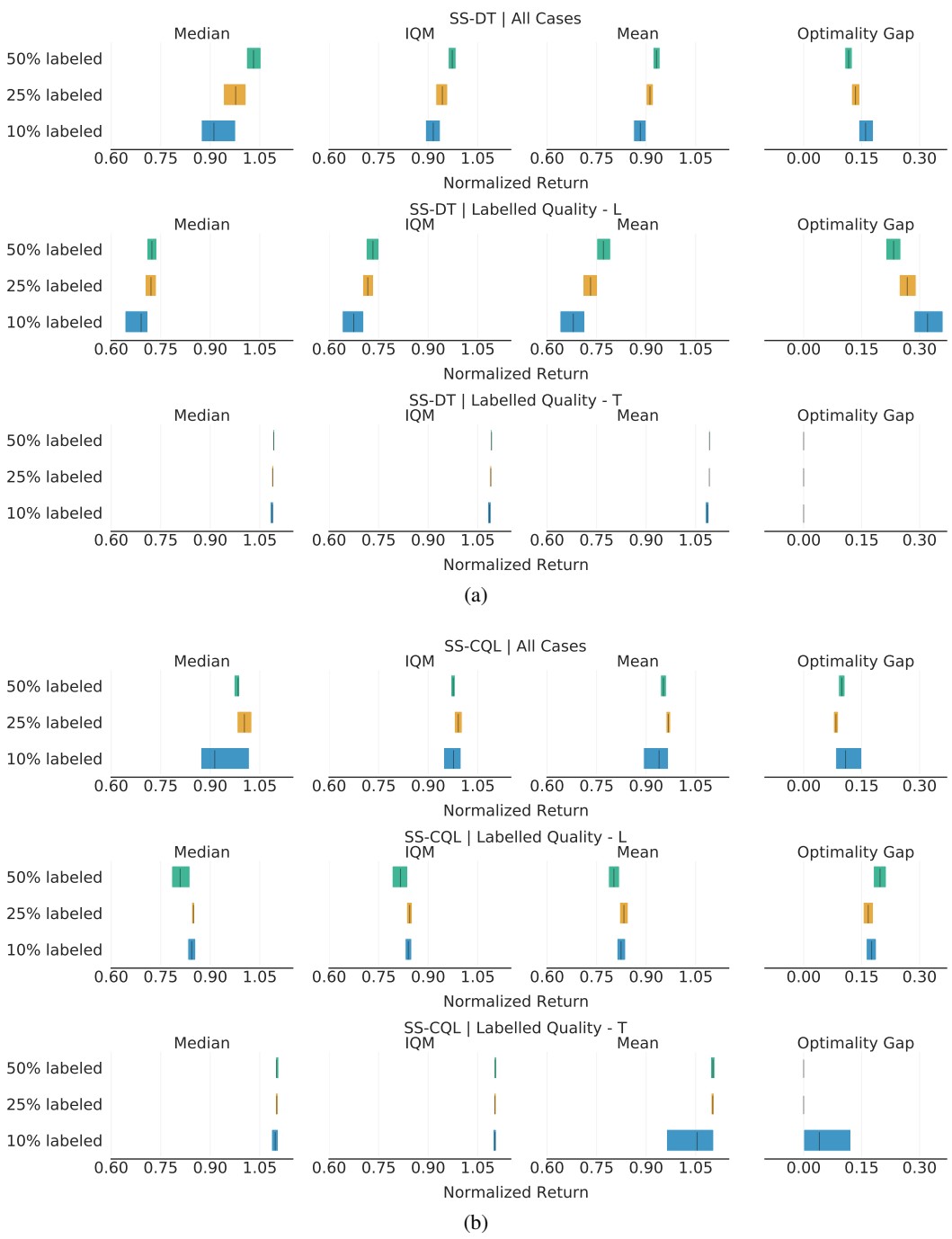

Figure D.2: The $95\%$ stratified bootstrap CIs of four statistics (the median, mean, interquartile mean, and the optimality gap) over the returns of `SS-DT` and `SS-CQL` across all the setups, when trained with different sizes of labelled trajectories. The number of unlabelled trajectories is $10\%$ of the total number of offline trajectories of corresponding dataset. We use $50000$ bootstrap replications to generate the CIs.

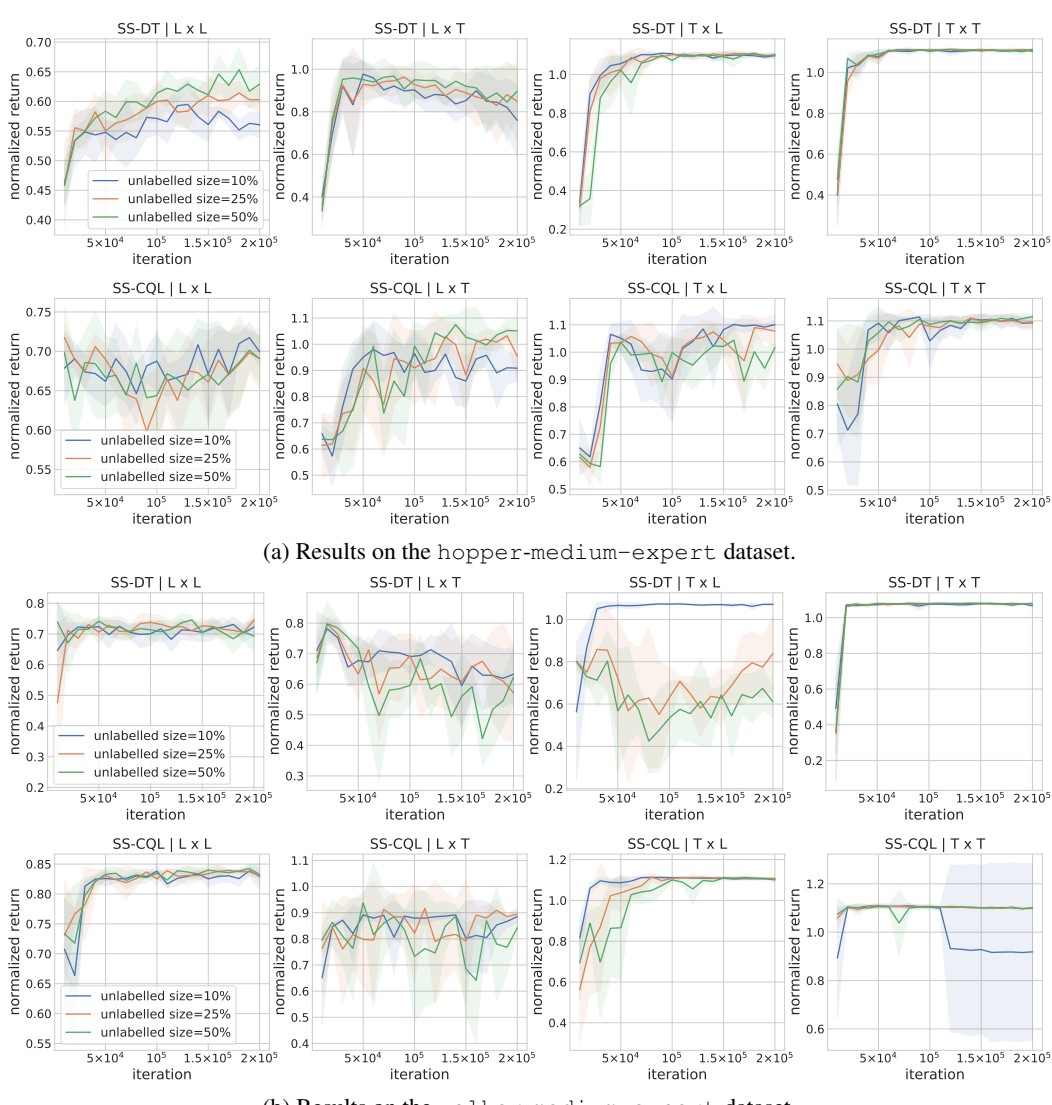

(a) Results on the `hopper-medium-expert` dataset.

(b) Results on the `walker-medium-expert` dataset.

Figure D.3: The return (average and standard deviation) of `SS-DT` and `SS-CQL` agents trained on the `medium-expert` datasets with different sizes of unlabelled data. The labelled data size is fixed to be 10% of the offline dataset size. Results aggregated over 5 instances with different seeds.

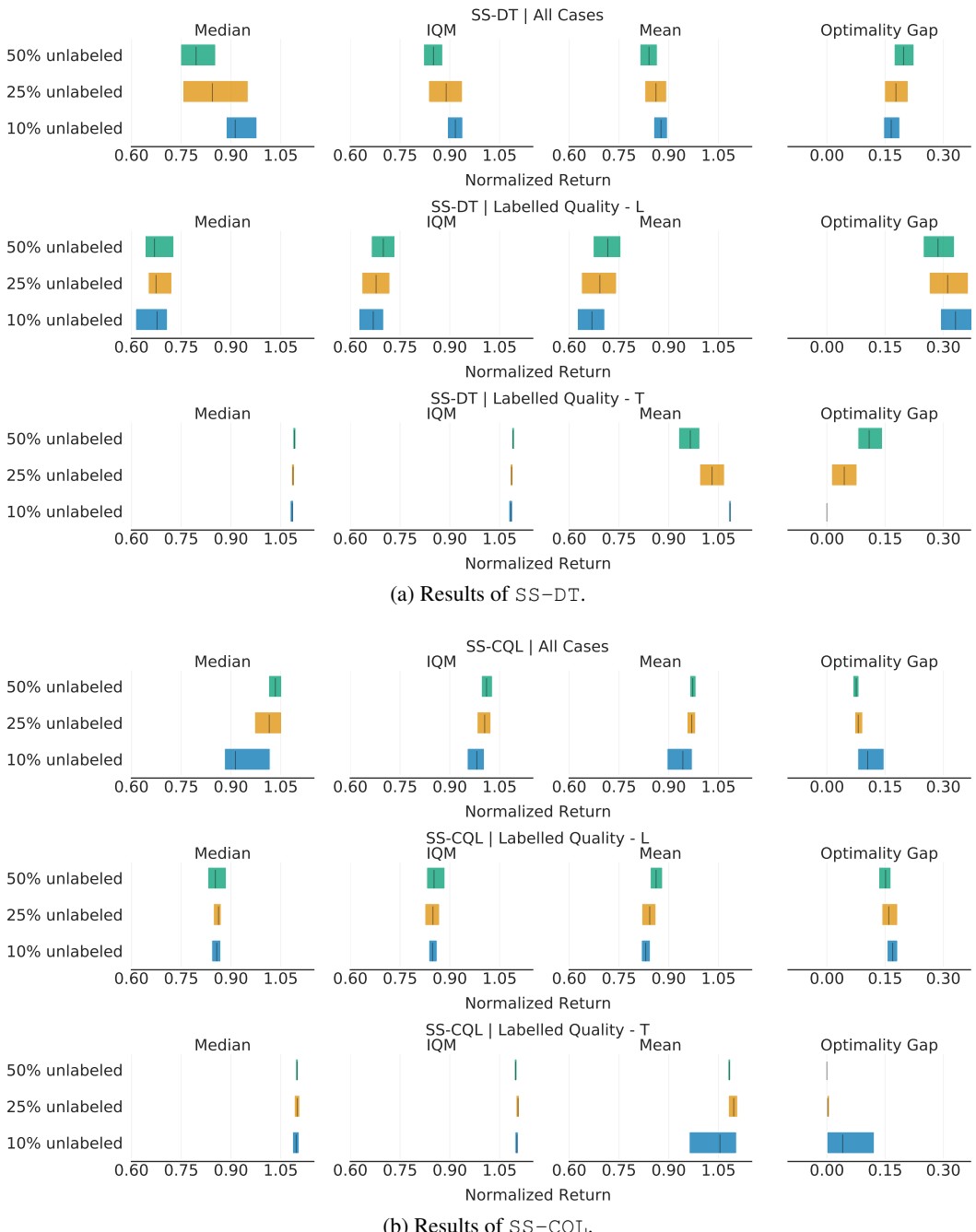

(a) Results of SS-DT.

(b) Results of SS-CQL.

Figure D.4: The $95\%$ stratified bootstrap CIs of four statistics (the median, mean, interquartile mean, and the optimality gap) of the returns of SS-DT and SS-CQL across all the setups, when trained with different sizes of unlabelled trajectories. The number of labelled trajectories is $10\%$ of the total number of offline trajectories of corresponding dataset. We use $50000$ bootstrap replications to generate the CIs.

# E    TRANSITION SIZE $k$ FOR THE MULTI-TRANSITION INVERSE DYNAMIC MODEL

## E.1    THEORY

Let $\beta$ denote the behaviour policy. When $k = 0$, the IDM is modeling

$$\mathbb{P}(a_t|s_{t+1}, s_t) = \frac{\mathbb{P}(a_t, s_{t+1}|s_t)}{\mathbb{P}(s_{t+1}|s_t)} = \frac{\mathbb{P}(s_{t+1}|a_t, s_t)\beta(a_t|s_t)}{\mathbb{P}(s_{t+1}|s_t)}. \tag{2}$$

For the cases where $k > 0$, w.l.o.g, we assume $k = 1$. The IDM is modeling

$$
\begin{aligned}
\mathbb{P}(a_t|s_{t+2}, s_{t+1}, s_t, s_{t-1}) &= \frac{\mathbb{P}(a_t, s_{t+2}, \ldots, s_{t-1})}{\mathbb{P}(s_{t+2}, \ldots, s_{t-1})} \\
&= \mathbb{P}(s_{t+1}|a_t, s_t, s_{t+2}, s_{t-1})\,\mathbb{P}(a_t|s_{t+2}, s_t, s_{t-1})/\mathbb{P}(s_{t+1}|s_{t+2}, s_t, s_{t-1}) \\
&= \mathbb{P}(s_{t+1}|a_t, s_t)\beta(a_t|s_t, s_{t-1})/\mathbb{P}(s_{t+1}|s_t, s_{t-1}),
\end{aligned}
\tag{3}
$$

where in the last line we used the fact that the policy $\beta$ can only generate actions based on previous states, the Markovian transition property $\mathbb{P}(s_{t+1}|a_t, s_t, s_{t+2}, s_{t-1}) = \mathbb{P}(s_{t+1}|a_t, s_t)$, and also the induced property $\mathbb{P}(s_{t+1}|s_t, s_{t-1}) = \mathbb{P}(s_{t+1}|s_{t+2}, s_t, s_{t-1})$.

If the behaviour policy $\beta$ is Markovian, we have that $\beta(a_t|s_t) = \beta(a_t|s_t, s_{t-1})$, and as a consequence

$$\mathbb{P}(a_t|s_{t+1}, s_t) = \mathbb{P}(a_t|s_{t+2}, s_{t+1}, s_t, s_{t-1}) \cdot C(s_{t+1}, s_t, s_{t-1}), \tag{4}$$

where $C = \frac{\mathbb{P}(s_{t+1}|s_t)}{\mathbb{P}(s_{t+1}|s_t, s_{t-1})}$ is independent of the action $a_t$. Therefore, the probabilities that the IDMs with $k = 0$ and $k = 1$ are modeling, are equivalent up to a state-only dependent scaling. The cases where $k \geqslant 2$ can be derived analogously.

In practice, the offline dataset might contain trajectories generated by multiple behaviour policies and it is unknown if any of them is Markovian. Therefore, choosing $k > 0$ allows us to take into account past information before timestep $t$. For future, we do not need anything beyond $s_{t+1}$ for a MDP, but our formulation is general purpose to account for POMDP as well, where both past and future partial observations might be needed to infer the action $a_t$. To summarize, choosing $k \geqslant 0$ is more general and have been shown to be favorable in the empirical experiments presented in next section.

## E.2    EMPIRICAL EXPERIMENTS

We train `SS-TD3BC`, `SS-CQL` and `SS-DT` with 3 IDM transition size: $k = 0, 1$ and 2 on the `hopper-medium-expert` dataset. We use the coupled setup described in Section 4.1, with 6 different values of $q$. Table E.1 reports the performance of those agents for each case.

In addition to the interquartile mean considered in Section 4.3, we also consider 3 other statistics of the return across all the setups: the mean, the median and the optimality gap. Figure 4.3 plots the 95% stratified bootstrap confidence intervals for all the four statistics, genereated by 50000 bootstrap replications.

|  |  | $q = 10$ | $q = 30$ | $q = 50$ | $q = 70$ | $q = 90$ | $q = 100$ | Average |
|---|---|---|---|---|---|---|---|---|
| SS-TD3BC | $k = 0$ | $0.81 \pm 0.12$ | $0.89 \pm 0.05$ | $0.93 \pm 0.05$ | $1.05 \pm 0.04$ | $1.03 \pm 0.06$ | $1.01 \pm 0.04$ | 0.95 |
|  | $k = 1$ | $0.93 \pm 0.07$ | $1.01 \pm 0.05$ | $0.86 \pm 0.06$ | $0.98 \pm 0.06$ | $1.03 \pm 0.06$ | $1.03 \pm 0.04$ | 0.98 |
|  | $k = 2$ | $0.80 \pm 0.12$ | $0.91 \pm 0.03$ | $0.93 \pm 0.05$ | $0.95 \pm 0.08$ | $1.01 \pm 0.06$ | $1.04 \pm 0.02$ | 0.94 |
| SS-CQL | $k = 0$ | $0.69 \pm 0.17$ | $0.69 \pm 0.15$ | $0.88 \pm 0.15$ | $1.04 \pm 0.04$ | $1.11 \pm 0.01$ | $1.10 \pm 0.03$ | 0.92 |
|  | $k = 1$ | $0.69 \pm 0.15$ | $0.90 \pm 0.05$ | $0.89 \pm 0.13$ | $1.03 \pm 0.07$ | $1.07 \pm 0.08$ | $1.11 \pm 0.01$ | 0.95 |
|  | $k = 2$ | $0.90 \pm 0.11$ | $0.90 \pm 0.09$ | $0.86 \pm 0.11$ | $1.08 \pm 0.05$ | $1.10 \pm 0.01$ | $1.11 \pm 0.01$ | 0.99 |
| SS-DT | $k = 0$ | $0.72 \pm 0.17$ | $0.75 \pm 0.20$ | $0.90 \pm 0.14$ | $1.06 \pm 0.04$ | $1.11 \pm 0.00$ | $1.11 \pm 0.01$ | 0.94 |
|  | $k = 1$ | $0.69 \pm 0.20$ | $0.94 \pm 0.07$ | $0.99 \pm 0.05$ | $1.05 \pm 0.04$ | $1.11 \pm 0.00$ | $1.11 \pm 0.00$ | 0.98 |
|  | $k = 2$ | $0.78 \pm 0.07$ | $0.89 \pm 0.08$ | $0.85 \pm 0.15$ | $1.05 \pm 0.02$ | $1.11 \pm 0.00$ | $1.11 \pm 0.00$ | 0.97 |

Table E.1: The return (average and standard deviation) of `SS-ORL` agents trained on the `hopper-medium-expert` dataset under the coupled setup, where the IDM is trained with 2 different values of $k$: 0, 1 and 2. Results aggregated over 5 training instances.

Figure E.1: The 95% stratified bootstrap CIs of four statistics (the median, mean, interquartile mean, and the optimality gap) of the returns obtained by SS-ORL agents, with different values of $k$.

# F  DATA AUGMENTATION STRATEGY

Following Lakshminarayanan et al. (2017), we train an ensemble of 3 independent IDMs on $\mathcal{T}_{\text{labelled}}$. Each individual IDM models the action as a diagonal Gaussian distribution (see Equation (1)) $\mathcal{N}(\mu_i, \Sigma_i)$, $i = 1, 2, 3$. The ensemble models the action using a equally weighted Gaussian mixture of these three distributions. We predict the action by the mixture's mean and predict the uncertainty by the mixture's variance; both can be written in close form.

We conduct experiments for SS-DT and SS-TD3BC, where we only add proxy-labelled data whose uncertainties are below $p\%$ to the final RL training dataset. Specifically, we test 4 values of $p$: $25, 50, 75$ and $95$.:w We compare the results with standard SS-DT and SS-TD3BC where all the proxy-labelled data are added into the final RL training dataset. We consider both the hopper-medium-expert and walker-medium-expert datasets. We use the coupled setup described in Section 4.1, where we consider 4 different values of $q$: $10, 30, 68, 75$ for hopper-medium-expert and $10, 30, 54, 60$ for walker-medium-expert.

Table F.1 reports the average return and standard deviation obtained by SS-DT and SS-TD3BC under different data augmentation strategies, when trained on the hopper-medium-expert dataset. The results on the walker-medium-expert dataset are reported in Table F.2. It is easy to see that uncertainty based data augmentation degrades the performance, compared with adding all the proxy-labelled data without filtering. Overall, the latter performs consistently well across different setups. Figure F.1 plots the 95% stratified bootstrap CIs for this experiments. All the statistics favor the *no filtering* strategy.

|  |  | hopper-medium-expert | | | | |
|---|---|---|---|---|---|---|
|  |  | $q = 10$ | $q = 30$ | $q = 68$ | $q = 75$ | Average |
|  | below 25% | $0.60 \pm 0.03$ | $0.62 \pm 0.02$ | $0.71 \pm 0.04$ | $0.86 \pm 0.02$ | 0.70 |
|  | below 50% | $0.62 \pm 0.02$ | $0.66 \pm 0.06$ | $0.76 \pm 0.04$ | $0.86 \pm 0.09$ | 0.72 |
| SS-TD3BC | below 75% | $0.70 \pm 0.06$ | $0.74 \pm 0.07$ | $0.84 \pm 0.06$ | $0.94 \pm 0.08$ | 0.80 |
|  | below 95% | $0.82 \pm 0.05$ | $0.82 \pm 0.09$ | $0.90 \pm 0.09$ | $0.96 \pm 0.06$ | 0.88 |
|  | no filtering | $0.80 \pm 0.07$ | $0.92 \pm 0.04$ | $0.91 \pm 0.06$ | $0.94 \pm 0.10$ | 0.89 |
|  | below 25% | $0.61 \pm 0.12$ | $0.62 \pm 0.05$ | $0.70 \pm 0.01$ | $0.95 \pm 0.13$ | 0.72 |
|  | below 50% | $0.60 \pm 0.14$ | $0.65 \pm 0.04$ | $0.69 \pm 0.02$ | $1.04 \pm 0.07$ | 0.75 |
| SS-DT | below 75% | $0.42 \pm 0.04$ | $0.63 \pm 0.15$ | $0.75 \pm 0.06$ | $1.04 \pm 0.04$ | 0.71 |
|  | below 95% | $0.51 \pm 0.16$ | $0.82 \pm 0.12$ | $0.85 \pm 0.05$ | $1.06 \pm 0.03$ | 0.81 |
|  | no filtering | $0.47 \pm 0.14$ | $0.71 \pm 0.14$ | $0.83 \pm 0.07$ | $1.06 \pm 0.03$ | 0.77 |

Table F.1: The return (average and standard deviation) of SS-ORL agents trained on the hopper-medium-expert dataset under the coupled setup, using different data augmentation strategies. Results aggregated over 5 training instances.

| | | \multicolumn{5}{c}{walker-medium-expert} |
|---|---|---|---|---|---|---|
| | | $q = 10$ | $q = 30$ | $q = 54$ | $q = 60$ | Average |
| SS-TD3BC | below 25% | $0.82 \pm 0.02$ | $0.82 \pm 0.01$ | $0.80 \pm 0.06$ | $1.04 \pm 0.06$ | 0.87 |
| | below 50% | $0.83 \pm 0.03$ | $0.84 \pm 0.02$ | $0.84 \pm 0.01$ | $1.02 \pm 0.09$ | 0.88 |
| | below 75% | $0.74 \pm 0.11$ | $0.86 \pm 0.01$ | $0.85 \pm 0.01$ | $1.04 \pm 0.07$ | 0.87 |
| | below 95% | $0.86 \pm 0.04$ | $0.88 \pm 0.01$ | $0.87 \pm 0.01$ | $1.10 \pm 0.01$ | 0.93 |
| | no filtering | $0.86 \pm 0.05$ | $0.86 \pm 0.03$ | $0.87 \pm 0.01$ | $1.10 \pm 0.01$ | 0.92 |
| SS-DT | below 25% | $0.69 \pm 0.04$ | $0.74 \pm 0.02$ | $0.70 \pm 0.03$ | $0.84 \pm 0.17$ | 0.74 |
| | below 50% | $0.67 \pm 0.03$ | $0.72 \pm 0.02$ | $0.73 \pm 0.03$ | $0.95 \pm 0.15$ | 0.77 |
| | below 75% | $0.71 \pm 0.03$ | $0.60 \pm 0.13$ | $0.73 \pm 0.03$ | $0.95 \pm 0.14$ | 0.74 |
| | below 95% | $0.73 \pm 0.08$ | $0.52 \pm 0.11$ | $0.58 \pm 0.15$ | $0.98 \pm 0.10$ | 0.70 |
| | no filtering | $0.79 \pm 0.05$ | $0.55 \pm 0.13$ | $0.69 \pm 0.08$ | $0.91 \pm 0.15$ | 0.74 |

Table F.2: The return (average and standard deviation) of `SS-ORL` agents trained on the `walker-medium-expert` dataset under the coupled setup, using different data augmentation strategies. Results aggregated over 5 training instances.

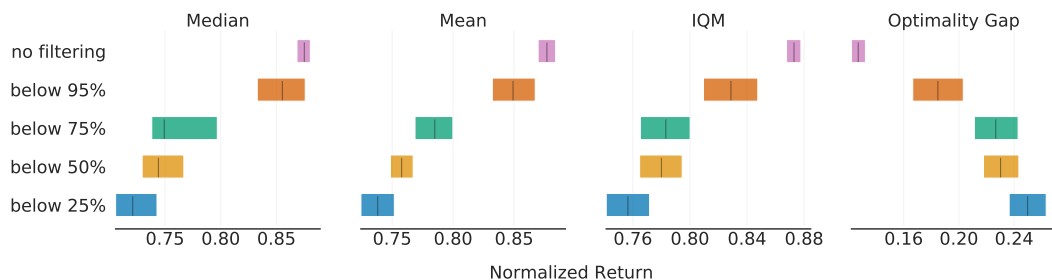

Figure F.1: The 95% stratified bootstrap CIs of four statistics (the median, mean, interquartile mean, and the optimality gap) of the returns obtained by `SS-ORL` agents, when combined with different data augmentation strategies.

# G  COMPARISON WITH GATO UNDER THE COUPLED SETUP

Inspired the multi-task and multi-modal generalist agent proposed by Reed et al. (2022), we consider a `GATO` type of variant of `DT` that can incorporate the unlabelled data into policy training.

`GATO` is trained on the labelled and unlabelled data simultaneously. The implementation details are:

- We form the same input sequence as `DT`, where we fill in zeros for the missing actions for unlabelled trajectories.
- For the labelled trajectories, `GATO` predicts the actions, states and rewards; for the unlabelled ones, `GATO` only predicts the states and rewards.
- We use the stochastic policy as in online decision transformer (Zheng et al., 2022) to predict the actions.
- We use deterministic predictors for the states and rewards, which are single linear layers built on top of the Transformer outputs.

Let $g_t = \sum_{t'=t}^{|\tau|} r_{t'}$ be the return-to-go of a trajectory $\tau$ at timestep $t$. Let $H_\theta^{\mathbf{P}_{\text{labelled}}}$ denotes the policy entropy included on the labelled data distribution. For simplicity, we assume the context length of `GATO` is 1. We refer the readers to Zheng et al. (2022) for the formulation with a general context length and more details. Equation (5) shows the training objective of `GATO`.

$$
\begin{aligned}
\min_{\theta} \quad & \mathbb{E}_{(a_t,s_t,r_t,g_t) \sim \mathbf{P}_{\text{labelled}}} \left\{ -\log \pi(a_t|s_t, g_t, \theta) + \lambda_s \|s_t - \widehat{s}_t(\theta)\|_2^2 + \lambda_r \|r_t - \widehat{r}_t(\theta)\|_2^2 \right\} \\
& + \mathbb{E}_{(s_t,r_t,g_t) \sim \mathbf{P}_{\text{unlabelled}}} \left\{ \lambda_s \|s_t - \widehat{s}_t(\theta)\|_2^2 + \lambda_r \|r_t - \widehat{r}_t(\theta)\|_2^2 \right\} \\
\text{s.t.} \quad & H_\theta^{\mathbf{P}_{\text{labelled}}}[a|s, g] \geqslant \nu
\end{aligned}
\tag{5}
$$

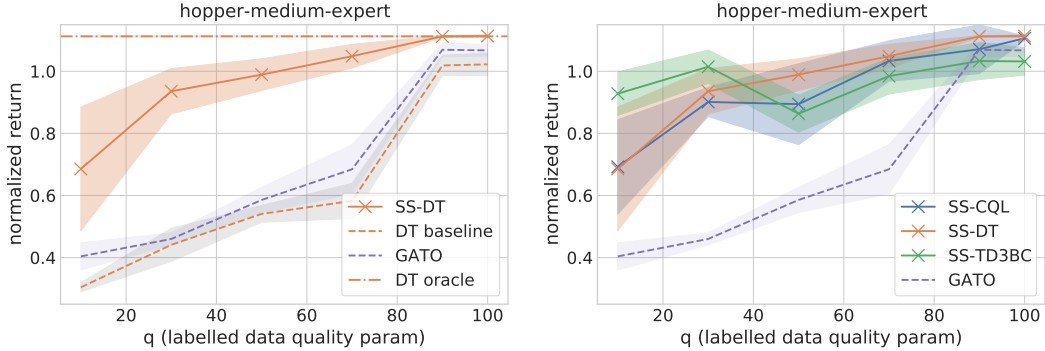

Figure G.1: (The performance of SS-ORL and GATO on the hopper-medium-expert dataset. For GATO, we use $\lambda_s = 0.01$ and $\lambda_r = 1.0$. (L) SS-DT significantly outperforms GATO, where GATO only slightly improves upon the baseline. (R) SS-CQL, SS-DT and SS-TD3BC all outperform GATO.

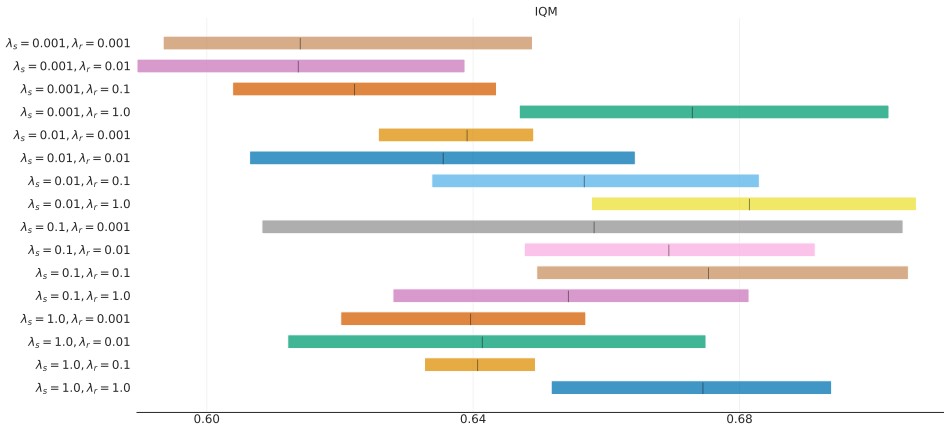

Figure G.2: The $95\%$ stratified bootstrap CIs of four statistics (the median, mean, interquartile mean, and the optimality gap) of the returns obtained by GATO agents, with different combinations of regularization parameters.

The constant $\nu$, $\lambda_s$ and $\lambda_r$ are prefixed hyper-parameters, where $\nu$ is the target policy entropy, and $\lambda_s$ and $\lambda_r$ are regularization parameters used to balance the losses for actions, states, and rewards. We use $\nu = -\dim(\mathcal{A})$ as for DT (see Appendix A). To choose the regularization parameters $\lambda_s$ and $\lambda_r$ for GATO, we test 16 combinations where $\lambda_s$ and $\lambda_r$ are $1.0, 0.1, 0.01$ and $0.001$ respectively. We run experiments as in Section 4.1 for $q = 10, 30, 50, 70, 90, 100$, and compute the confidence intervals for the aggregated results. Figure G.2 shows that $\lambda_s = 0.01$ and $\lambda_r = 0.1$ yield the best performance.

Figure G.1 compares the performance of GATO (with $\lambda_s = 0.01$ and $\lambda_r = 0.1$) and SS-ORL agents. It is clear that SS-ORL agents outperform GATO.

## H   PERFORMANCE GAP OF SS-ORL AGENTS

For a chosen offline RL method, the relative performance gap between the corresponding SS-ORL and oracle agents illustrates how sensitive this offline RL is to missing actions:

$$\frac{\text{Oracle-ORL} - \text{SS-ORL}}{\text{Oracle-ORL}}. \tag{6}$$

We consider the coupled setup as in Section 4.1. For each of the 9 datasets (hopper, walker, halfcheetah with medium-expert, medium, and medium-replay datasets), we compute the relative performance gap for SS-CQL, SS-DT and SS-TD3BC, trained with 6 different values of $q$: $10, 30, 50, 70, 90$ and $100$. Table H.1 reports the aggregate results over 5

seeds. On average, `SS-CQL` and `SS-TD3BC` have smaller relative performance gap, suggesting that CQL and TD3BC are less sensitive to the missing actions.

| method | hopper-me | walker2d-me | hc-me | hopper-m | walker2d-m | hc-m | hopper-mr | walker2d-mr | hc-mr | Average |
|---|---|---|---|---|---|---|---|---|---|---|
| SS-CQL | 0.147 | 0.114 | 0.062 | 0.078 | 0.077 | 0.003 | 0.388 | 0.379 | 0.106 | 0.150 |
| SS-TD3BC | 0.046 | 0.094 | 0.104 | 0 | 0.065 | 0.001 | 0.327 | 0.412 | 0.057 | 0.123 |
| SS-DT | 0.119 | 0.167 | 0.0002 | 0.016 | 0.039 | 0.003 | 0.399 | 0.554 | 0.109 | 0.156 |

Table H.1: The relative performance gap of `SS-CQL`, `SS-TD3BC`, and `SS-DT`.

