# OpenReview forum: "Semi-Supervised Offline Reinforcement Learning with Action-Free Trajectories"
_ICLR.cc/2023/Conference — Submitted to ICLR 2023_

### Official Review · Reviewer_omLk · 2022-10-14

**Confidence:** 5
**Correctness:** 3
**Technical Novelty And Significance:** 1
**Empirical Novelty And Significance:** 3
**Recommendation:** 5

**Clarity, Quality, Novelty And Reproducibility:**

The paper is easy to follow in general and the writing is clear.
The quality of empirical evaluations is fine.
The methodology originality is low but the empirical study on the offline RL setting is novel.

**Strength And Weaknesses:**

Strength:
1. The setting is well-motivated and has decent potential for real-world applications.
2. The empirical studies are thorough and in particular, ablation over the quality of data is well-done.

Weaknesses:
1. The proposed method is quite standard in online settings. The claimed novelty (multiple transitions as input to the inverse model) is more of technical detail and the reason why it is helpful for the Markovian setting is not explained properly.
2. Most of the experiment results in the paper are based on the medium-expert dataset of d4rl gym-locomotion. This raises questions about whether the conclusion can be generalised to the setting with more diverse data. For medium-expert, the behaviour policy is basically a mixture of two policies (medium level and expert level). If the trajectories are split into multiple groups according to the returns, it's likely that the root cause of the varied returns is initial states rather than the quality of the policy. One clue is the experiments on medium-replay data in Figure C.2 show for hopper and walker2d, the quality of the unlabelled data plays a much more important role than the case in the medium and medium expert.

Minor issues
1. Is "label" a proper word to replace action? My impression is unlabelled data in offline RL is more about reward-missing data. The intuition behind that is humans can label rewards to trajectories easily because it's a scale but "labelling" actions seems like very difficult.
2. There are some obvious grammatical errors in the paper: e.g. "we are mainly interested in the case **only** a **significant majority** of the trajectories in the offline trajectories are **unlabeled**." "How can we utilize the unlabelled data for improving **the performance offline RL algorithms**?"


**Summary Of The Paper:**

The paper provides an empirical study about a setting where a portion of the offline RL dataset doesn't include actions. To exploit action-missing data, this work proposes to learn an inverse dynamics model on data with actions to generate proxy actions from state transitions. A set of empirical studies on d4rl gym-locomotion control are provided to give insights into how such semi-supervised learning will be helpful for the performance of the final policy. The ablation studies show the proposed semi-supervised method is particularly helpful when the action-missing data is of high quality and labelled data is of lower quality.

**Summary Of The Review:**

The paper studies an interesting setting for offline RL which has significant practical value. The study of different scenarios of data availability and the quality of the data is a timely topic for offline RL research.
However, my main concern about this paper is it's not very informative. The whole paper looks more like a technical report about applying the inverse dynamics model to the d4rl dataset and I'm struggling to figure out what's the key takeaway.
The authors summarised three of their key findings in the introduction:
1. One interesting claim of the paper is SS-ORL agent works well in the setting with lower-quality labelled and high-quality unlabelled data. But I'm not sure if such a claim can generalize to the setting when the policies are more diverse than a single or a mixture of two unimodal policies, as I argued in the second point of the weakness.
2. "When the labelled data quality is high, utilizing unlabelled data does not bring significant benefits." This might just reveal that most of the data from the d4rl gym-locomotion are redundant because of the low diversity of the policy?
3. "CQL and TD3BC are less sensitive to the action labels compared to DT." This is not very obvious to me by just looking at Figure 4.1 and Figure 4.2. It would be better to have quantitative results if that's true.

Given the reasons above, I cannot recommend acceptance at the current stage. But I may change my mind if the authors or other reviewers can convince me about point 1, or remind me why point 2 and 3 are significant enough to accept the paper.

---

> ### Author Response · Authors · 2022-11-14
> **Response to Reviewer omLK**
>
> We thank the reviewer for identifying our strengths in proposing a well-motivated setting and  conducting a thorough empirical studies. We also thank you for the detailed comments. Please see our response below.
>
> * **novelty claim** Our novelty claims extend beyond the use of a multi-step dynamics model. In particular, we believe novelty should not be restricted to simply modeling contributions but also include analysis of existing protocols and their functioning under different operating conditions. To this end, we believe our key contribution is a data-centric analysis of different offline RL methods
> (value-based/cloning-based) under different assumptions on the quality of the labelled and unlabelled datasets. Moreover, we agree SS-ORL is simple and straightforward. We in fact believe that to be a strength of our approach and we did consider and compare SS-ORL with more complicated designs and have found that SS-ORL consistently outperforms them:
>     1. We considered only adding proxy-labelled trajectories/transitions with only low uncertainty into the final dataset for training offline agents -- inspired by the popular semi-supervised learning strategy “self-training”, see Section 4.3, Data Augmentation Strategy and Section 3.2 Remarks. SS-ORL consistently outperforms this variant with various uncertainty thresholds, see Figure 4.5.
>     2. In the updated version, we have also included a comparison with GATO [1], a recently proposed Transformer-based offline RL architecture that can train on both action-free trajectories and complete trajectories simultaneously. SS-DT outperforms GATO with significance, see Appendix G.
>
>     Based on the empirical results, we think the simplicity of SS-ORL is an asset as it makes it more scalable and easy to implement in other scenarios.
>
>     [1] A generalist agent. Reed et al. 2022
>
> * **generalization to settings with more diverse return distributions** (weakness 2 and summary 1) To clarify, the x-axis in Figure C.2 shows the quality of the labelled data. If we understand the question correctly, you mean for the hopper/walker2d-medium-replay datasets, when the value of $q$ increases, SS-ORL does not catch up with the oracle as quickly as on the medium and medium-expert datasets. You are thinking if this is because the return distributions of the medium-replay datasets are not bimodal like the returns distributions of medium-expert datasets.
>
>     Our intuition is that  the hopper/walker-medium-replay datasets’ return distributions concentrate on extremely low values --  the density function seems exponentially decaying as the return increases, see Appendix B for the density plots. More specifically, in our experiments, the labelled trajectories for those two datasets have average return small than 0.1 even when $q=70$. In contrast, the return distributions of the other datasets concentrate on larger values, where increasing the value of $q$ will greatly change the returns of labelled trajectories. We have updated Appendix C to include experiments on a subsampled dataset -- where the return distribution is wide. In this experiment, SS-ORL still can catch up with with the oracle when $q$ is small.
>
> * **wording (the use of label for action)** Indeed, there are papers that refer to “reward” when they use “label”.  In the (semi-)supervised learning literature, “label” refers to the target variable. Both rewards and actions are training signals that can serve as targets. In our context, we use the inverse dynamic model to predict the missing actions, so we think it is intuitive to use “label” for the actions here. We hope from our writing, it is easy for the reader to pick up this terminology and will add further clarifications.
>
> * **grammatical errors** Thank you for highlighting the grammar typos – we have fixed them in the updated version.
>
> * **the data from the d4rl gym-locomotion are redundant**  We think you are referring to Figure 4.2. We used hopper-medium-expert dataset in this illustrative example. The return distribution of this dataset is quite wide and sufficiently diverse, as illustrated in Appendix B, Figure B.1. When the labelled trajectories are sampled from the high quality group (top ⅓ ones, in this example they are expert trajectories), training offline RL methods on them directly already achieves expert performance (see Figure 4.2), so that adding more unlabelled trajectories does not further improve the performance.

---

> > ### Author Response · Authors · 2022-11-14
> > **Response to Reviewer omLK (continued)**
> >
> > * **CQL and TD3BC are less sensitive to the missing actions** Thanks for the suggestion, we agree presenting quantitative numbers will demonstrate this better.  Specifically, we can quantify the relative performance gap between SS-ORL and the corresponding oracle agent to  illustrate the sensitivity of an offline RL method to the missing actions:  (Oracle-ORL - SS-ORL) / Oracle-ORL. For each of the 9 datasets (hopper/walker2d/halfcheetah x medium-expert/medium/medium-replay), we compute the relative gap for SS-ORL trained with 6 different values of $q$: 10, 30, 50, 70, 90 and 100. The following table reports the aggregate results:
> > method| hopper-me| walker2d-me | hc-me | hopper-m| walker2d-m| hc-m | hopper-mr| walker2d-mr| hc-mr| Average
> > -------|--------|-------|-------|-------|-------|-------|------|-------|-------|-------
> > SS-CQL  | 0.147	| 0.114	| 0.062 | 	0.078	| 0.077	| 0.003| 	0.388| 	0.379| 	0.106| 	0.150
> > SS-TD3BC  | 0.046 |	0.094	|0.104	|0|	0.065|	0.001|	0.327|	0.412|	0.057|	0.123
> > SS-DT  | 0.119 |	0.167|	0.0002 |	0.016 |	0.039 |	0.003 |	0.399 |	0.554 |	0.109 |	0.156
> >
> >     On average the relative performance gap of SS-CQL and SS-TD3BC are smaller than SS-DT, suggesting CQL and TD3BC are less sensitive to the missing actions. We have included this analysis in Appendix H.
> >
> > Thank you for reviewing this paper and discuss our work. We hope the response above and the updated manuscript address your concerns. Please let us know if you have any other questions.

---

> > > ### Comment · Reviewer_omLk · 2022-11-23
> > > **Response to the Rebuttal**
> > >
> > > I'd like to thank the authors for their rebuttal.
> > >
> > > ### Novelty and Claim
> > > I agree that SS-ORL is simple and that's indeed a major strength compared to more complicated methods. On the other hand, the baseline "adding proxy-labelled trajectories/transitions with only low uncertainty" doesn't makes sense, at least to me. I'm also confused by why the authors are comparing SS-ORL to it. My understanding is the main point of Gato is to train the Transformer on multi-modality data otherwise it's just a Trajectory Transformer with one-step imitation learning. But the authors did was something completely irrelevant, it's more like a DT that also models state transition, which was commented out in the DT implementation. Gato has discretized state/action space to help it model the distribution of actions and states (rather than deterministic, or expected state/actions). Gato is also not conditioned on the returns.
> > >
> > > ### generalization to settings with more diverse return distributions
> > > When talking about diversity I actually mean the diversity of the policy rather than the returns. One can check the policy diversity by using the [D4RL data visualizer](https://github.com/Farama-Foundation/D4RL/blob/master/scripts/visualize_dataset.py). Datasets with the name "xxx-medium" or "xxx-expert" are generated by a single modal (gaussian) policy and generated behaviours are therefore similar. The different returns are caused by state initialization and the gaussian noise. "xxx-medium-expert" is just a mixture of the medium and expert datasets. "xxx-medium-replay" are trajectories taken from the replay buffer which have much higher diversity because the weights of the network are kept updating when generating the dataset.
> > >
> > > ### the data from the d4rl gym-locomotion are redundant
> > > To clarify, I mean even the top 1/3 of the data might have good coverage for the offline RL algorithm to work. One piece of evidence is in the DT paper even 10% BC works quite well on d4rl gym-locomotion. To really verify if the action labelling is helpful, one can, for example, just keep 1% or even a handful of the high-quality trajectories.
> > >
> > > ### CQL and TD3BC are less sensitive to the missing actions
> > > Thanks for the qualitative results. It seems like TD3BC is indeed less sensitive to the CQL/DT but the difference between CQL and DT is marginal, so I don't think the evidence is strong enough to make the claim interesting.

---

> > > > ### Author Response · Authors · 2022-12-02
> > > > **Response to Reviewer omLK**
> > > >
> > > > Thank you for the feedback! Regarding your questions:
> > > >
> > > > * **Why we thought about comparing to “uncertainty based data augmentation” and GATO**
> > > >     1. **uncertainty based data augmentation** The reason why we compare SS-ORL with “uncertainty based data augmentation” is that this is a well-known approach for semi-supervised learning called self-training (Farlick, 1967).  It iterates the following 3 steps: 1) train the predictor 2) annotate the unlabelled data 3) add a subset of annotated data with low uncertainty into the training set. For SS-ORL, the annotation process is to train the IDM and obtain proxy-actions of the unlabelled trajectories. However, our end goal is to obtain a downstream policy with improved performance, rather than simply labelling the actions. Please see the discussions in Section 3.2. This leads to our experiment in Section 4.3, where we only add proxy-labelled trajectories with low uncertainties and see if it improves performance. However, we found that this underperforms SS-ORL. Moreover, training with multiple rounds is computationally expensive. SS-ORL does not retrain the IDM but directly moves to the next state (policy training), which is a win both for effectiveness and efficiency.
> > > >
> > > >     2. **GATO** The central question we are concerned with, and the underlying theme of this paper, is how to utilize unlabelled trajectories to improve policy learning. The goal of this experiment is to test a different way of leveraging unlabelled trajectories: specifically, we try to predict the next state tokens using the same transformer-based architecture as the one used to predict actions. This can be viewed as an auxiliary task for the model which provides additional supervision signals. This is similar to GATO in the sense that GATO also predicts different types of tokens with the same shared architecture. However, calling this GATO may have been confusing, and we will adopt a different name for this baseline in the revised paper.
> > > >
> > > > * **Generalization** Thank you for the clarification. We think you are concerned with that when we split the trajectories based on the return, the low-return trajectories are also generated by the relatively high-quality policy and the differences in returns are caused by different initial states. To exclude the possibility that different returns are caused by the initial states or random noise, we have visualized the PCA projections of the initial states of all the trajectories in hopper/walker2d's medium/medium-replay/medium-expert datasets. For all these datasets, the initial states are well mixed for trajectories with low/high returns, without any cluster/group structures. The data points look like (isotropic) Gaussian distributed. This provides strong evidence that the differences of returns stem from the qualities of policies, rather than environmental randomness. In the updated version, we will include this analysis.
> > > >
> > > > * **the data from the d4rl gym-locomotion are redundant** Following your suggestions, we run SS-ORL with 1%, 3%, 5% and 8% labelled trajectories on the walker2d-medium-expert-v2 dataset, using the same setting as before. We have found that:
> > > >     1. As the quality parameter q increases, the baseline (offline RL trained on the labelled data only) could not match with the oracle
> > > >     2. SS-ORL is able to match with the oracle performance, with similar trends as we presented in the main paper: it consistently outperforms the baseline, and only requires relatively low quality data.
> > > >
> > > > For simplicity, we present the results of  the 1% case when q=100 in the following table:
> > > >   method       | oracle | ss-orl | baseline |
> > > > -------|--------|-------|-------|
> > > > CQL | 1.103 | 1.101 | 0.919
> > > > DT | 1.073 |1.071 | 0.716
> > > > TD3BC | 1.102 | 1.107 | 1.014
> > > > The results provide strong evidence for the effectiveness of SS-ORL. We will include the latest results in the updated version for more detailed discussion.
> > > >
> > > > * **sensitivity to the missing actions** Thank you for the suggestion. We will update the claim to be that we have found TD3BC (the hybrid approach) is less sensitive to the missing actions.
> > > >
> > > > Please let us know your thoughts and/or any other questions you'd like to discuss.

---

### Official Review · Reviewer_az7q · 2022-10-25

**Confidence:** 2
**Correctness:** 3
**Technical Novelty And Significance:** 2
**Empirical Novelty And Significance:** 2
**Recommendation:** 3

**Clarity, Quality, Novelty And Reproducibility:**

This paper is clearly written, with some minor typos throughout. Overall, my main concern is with the novelty and quality of the proposed algorithm. The results should be reproducible.

**Strength And Weaknesses:**

Weaknesses:
- My main concern is with the technical contribution of this paper. Considering the inverse dynamics model as the crucial technical novelty, I am not sure the method given in Eq. 1 is the best, and at the very least, some further evaluations and arguments for its design choice is needed. For one, the covariate matrix with k>0 would seem to break the markov property of the RL setting. Furthermore, it is not justified why this choice was made beyond the empirical validation, which is not enough either in my opinion. I am not convinced that this extension of IDM significantly contributes to the overall goal of taking advantage of semi-supervision, as compared to the fully unsupervised case. In particular, it would have been interesting to see whether semi-supervision in this regime would help when the data split is within trajectories and not between trajectories.
- Along the lines of the above, I think an analysis of why only a single labelling round was used instead of the conventional self-training paradigm of retraining per round could have been included to make the paper stronger. In particular, I assume it could be possible to provide a more detailed analysis of how well the inverse dynamics model learns when compared to the ground truth data on the log-likelihood of the multivariate Gaussians used to estimate them since you have access to those data labels.
- As it stands, this paper's main contributions are a careful study of the different considerations one should take when doing semi-supervised offline RL. It has a good experimental validation of how performant value-based and BC methods. However, it does not make enough technical contribution to take advantage of the specific literature in the semi-supervised learning setting or does not justify its design choices well.

**Summary Of The Paper:**

This paper introduces a method for semi-supervised learning in the offline RL setting where the unlabelled part of the dataset consists of action-free state trajectories and the labelled part consists of the full trajectories. They use an inverse dynamics model to learn actions that give rise to state transitions and use the learned model to inject labels for unlabelled data and perform offline learning using classic model-free offline RL algorithms such as CQL.

**Summary Of The Review:**

Overall, I lean towards rejecting this paper. My reasoning is that while it is tackling a notable practical problem in the RL setting, there is not enough technical contribution nor is it polished enough in its exposition of the design choices for me to recommend acceptance.

---

> ### Author Response · Authors · 2022-11-14
> **Response to Reviewer az7q**
>
> We thank the reviewer for identifying our contribution in proposing a notable and practically motivated problem in RL. We also thank you for the insightful comments, and please see our response below.
>
> * **novelty and contribution** We believe one important novelty of our work is the proposed setup bridges the semi-supervised learning and RL areas. As the reviewer agreed on, our key contribution is the rigorous data-centric analysis of offline RL under the semi-supervised setup.
> While developing an outstanding semi-supervised learning algorithm is our primary focus, as our goal extends beyond learning the missing actions but a downstream offline RL agent, we did consider other more complex designs, which all underperform SS-ORL:
>
>     1. We considered only adding proxy-labelled trajectories/transitions with only low uncertainty into the final dataset for training offline agents -- which can be viewed as one step of “self-training”, see Section 4.3, Data Augmentation Strategy and Section 3.2 Remarks. SS-ORL consistently outperforms this variant with various uncertainty thresholds, see Figure 4.5. Given the undesired preliminary results, we did not extend SS-ORL to multiple labelling rounds.
>
>     2. In the updated version, we have also included a comparison with GATO [1], a recently proposed Transformer-based offline RL architecture that can train on both action-free trajectories and complete trajectories simultaneously. SS-DT significantly outperforms GATO, see Appendix G.
>
> Given the favorable empirical performance of SS-ORL, we believe the simplicity of SS-ORL is an additional strength: it makes the algorithm scalable and easy to implement.
>
> * **theoretical backup for the multi-transition IDM**  We have updated Appendix E to include the theoretical analysis. We can show that under the Markovian transition property: $s_{t+1} \sim P( \cdot |a_t, s_t)$, if there is a single behaviour policy $\beta$ of the dataset and $\beta$ is Markovian, i.e. $a_t \sim \beta(\cdot | s_t)$, the IDMs with $k >0$ and $k>1$ will model the same probability up to a state-only dependent scaling.  In practice, the dataset can be generated by multiple behaviour policies and it is unknown that if any of them is Markovian, the choice $k>0$ is more general and shown to be favored by our empirical experiments. Besides, our formulation can account for POMDPs where both past and future partial observations might be needed to infer the current action.
>
> * **data split within the trajectory**
> The practical motivation of our paper suggests the setup where the trajectories either contain or miss actions. We agree the within trajectory data split is interesting, but not directly related to our primary focus in this paper.
>
> [1] A generalist agent. Reed et al. 2022
>
> We hope our response and the updated manuscript address your concerns. Please let us know if you have any other questions. Thank you.

---

### Official Review · Reviewer_Mz9R · 2022-10-25

**Confidence:** 3
**Clarity, Quality, Novelty And Reproducibility:** The detailed dataset and code of the …
**Correctness:** 3
**Technical Novelty And Significance:** 3
**Empirical Novelty And Significance:** 3
**Recommendation:** 6

**Strength And Weaknesses:**

The paper is well-written and easy to follow. The setting is novel and meaningful since there are a lot of unlabeled data like videos in real-world scenarios.  The experiments are designed carefully to illustrate the influence of data with different qualities. For the weakness, Iwould like to ask some questions:
1. For IDM, the length of the input, k, could be changed, and k=1 in the paper. So the input of the model is four states (s
_t-1,..., s_t+1). How does the model encode these states?
2.  The experiments only use the expert dataset, which means most trajectories are good. D4RL also provides random and medium-level dataset. As your claim in the paper, the quality of the data has a huge influence on the performance. Is there any analysis based on these data? Could we say that the method also needs high-quality data for both labeled and unlabeled data to achieve good performance?

**Summary Of The Paper:**

This paper introduces a new, practically motivated semi-supervised setting, where the agent can access both labeled trajectories and unlabelled trajectories that do not include the actions of the trajectories. A model is trained to give actions for unlabeled data and then the offline RL method is trained by the whole dataset. Experiments based on D4RL dataset show the good performance of the proposed method.

**Summary Of The Review:**

See the above-mentioned questions.

---

> ### Author Response · Authors · 2022-11-14
> **Response to Reviewer Mz9R**
>
> Thank you for identifying our strengths in proposing a novel and meaning setting and conducting carefully designed experiments.  Below we provide responses to your questions. We hope that we could resolve your remaining concerns.
>
> * **how does the IDM encode multiple states** We concatenate the four states into a single vector. The concatenated vector is then fed into two MLPs, which predicts the mean and covariance matrix of the action distribution of $a_t$, respectively. Please see Section 4.1, Inverse Dynamic Model.
>
> * **Experiments on medium-level datasets and whether we need both high-quality labelled and unlabelled data**
> We did run experiments on the medium and medium-replay datasets. Due to the space limit, we deferred the results to Appendix C. We mainly focus on the medium-expert dataset in the main paper, because it is intuitive to illustrate our motivations.
> The conclusions and observations on those datasets are consistent with those made on medium-expert dataset:
>
>     1. Compared with the baseline agent (trained on the labelled trajectories only), SS-ORL improves the performance when the unlabelled data quality >= labelled data quality. (See Figure 4.1, C.1 and C.2 for the results on the coupled setup, and Figure 4.2 for the decoupled setup for the ablation study.) Note that the baseline performance is a lower bound of SS-ORL performance.
>
>     2. Compared with the oracle agent (trained on the fully labelled offline dataset),  SS-ORL performance is close to or even matches the oracle performance, often with a small fraction of low quality labelled data and high quality unlabelled data.  (See Figure 4.1, C.1, C.2 and 4.2 again.) Note that the oracle performance is an upper bound.
>
>     Here we are measuring quality relatively to the maximum return in a given offline dataset.
>
> Please let us known if you have any other questions. Thanks for taking your time to discuss our work.

---

### Author Response · Authors · 2022-11-14
**General Response (PDF Updated)**

We thank the reviewers for the constructive feedback. We have updated the manuscript to incorporate the suggestions. Please see our revision, where we have highlighted modifications in orange to ease reading.

The updated manuscript includes the following additional analysis and experiments to demonstrate and ablate the effectiveness of SS-ORL:
1. Theoretical analysis of the multi-transition inverse dynamic model. (Appendix E.1)
2. Comparison with GATO [1], a recently proposed multi-task sequence modeling architecture for offline RL that can be trained on the labelled and unlabelled trajectories simultaneously. We show that the performance of SS-ORL significantly outperforms GATO, demonstrating the effectiveness our simple approach compared to more sophisticated designs. (Appendix G)
3. Experiments of SS-ORL a subsampled dataset where the return distribution is wide. The results demonstrate that our claim extends to more diverse datasets. (Appendix C)
4. Quantitative measure of the performance gap of SS-ORL to the oracle agents. The results demonstrate that CQL & TD3BC is less sensitive to the missing actions than DT. (Appendix H)

We will respond to each reviewer in detail below, and we hope our response addresses your concerns. Please let us know if you have any other questions or comments.

[1] A generalist agent. Reed et al. 2022

---

### Author Response · Authors · 2022-11-16
**Rebuttal Reminder**

Thank you again for your review. It has been a while since we post our response, and we hope you have had a chance to read our response. We believe our reply addresses the weaknesses and questions raised by this review. Since the discussion period is short and drawing to a close soon, are there further questions or concerns we should discuss? Please let us know if there are anything we can help.

---

### Author Response · Authors · 2022-11-18
**Reminder again...**

Dear reviewers,

We sincerely hope you can spend time reading our response and let us know your thoughts.

Thanks,

Authors of Paper 2369

---

### Decision · Program_Chairs · 2023-01-20

**Decision:**

Reject

**Justification For Why Not Higher Score:**

The empirical results are still not convincing enough, especially in the variety of experiment settings, datasets, and return distributions.  The rebuttals have alleviated some concerns, but the paper still needs a new round a revision and review.

**Justification For Why Not Lower Score:**

N/A

**Metareview: Summary, Strengths And Weaknesses:**

This paper addresses offline reinforcement learning where two sets of trajectories are available: one with fully labeled state-action-reward, and one only with state-reward.  The idea is to learn an inverse-dynamic model from the labeled trajectories so that the unlabeled trajectories can be imputed.  By applying standard offline RL algorithms to the new trajectory set, superior empirical performance is demonstrated on some D4RL datasets.

Overall, the paper is well written, and addresses an important problem.  However, the empirical results are still not convincing enough, especially in the variety of experiment settings, datasets, and return distributions.  The rebuttals have alleviated some concerns, but the paper still needs a new round a revision and review.